# Generalizability assessment of AI models across hospitals in a low-middle and high income country

Jenny Yang [1] ✉, Nguyen Thanh Dung[2], Pham Ngoc Thach[3], Nguyen Thanh Phong[2], Vu Dinh Phu[3], Khiem Dong Phu[3], Lam Minh Yen[4], Doan Bui Xuan Thy [4], Andrew A. S. Soltan [1,5,6], Louise Thwaites[4,7,9] & David A. Clifton [1,8,9]

The integration of artificial intelligence (AI) into healthcare systems within low-middle income countries (LMICs) has emerged as a central focus for various initiatives aiming to improve healthcare access and delivery quality. In contrast to high-income countries (HICs), which often possess the resources and infrastructure to adopt innovative healthcare technologies, LMICs confront resource limitations such as insufficient funding, outdated infrastructure, limited digital data, and a shortage of technical expertise. Consequently, many algorithms initially trained on data from non-LMIC settings are now being employed in LMIC contexts. However, the effectiveness of these systems in LMICs can be compromised when the unique local contexts and requirements are not adequately considered. In this study, we evaluate the feasibility of utilizing models developed in the United Kingdom (a HIC) within hospitals in Vietnam (a LMIC). Consequently, we present and discuss practical methodologies aimed at improving model performance, emphasizing the critical importance of tailoring solutions to the distinct healthcare systems found in LMICs. Our findings emphasize the necessity for collaborative initiatives and solutions that are sensitive to the local context in order to effectively tackle the healthcare challenges that are unique to these regions.

As the field of artificial intelligence (AI) progresses, the integration of AI into healthcare systems presents a remarkable opportunity to revolutionize the delivery of healthcare, foster innovation and discovery, and ultimately enhance patient care and treatment outcomes on a global level. Nevertheless, while many high income countries may be well-prepared to develop and adopt these innovative technologies, the implementation of healthcare AI in low-middle income country (LMIC) settings poses distinctive challenges in comparison to high income country (HIC) settings.

LMIC hospitals often face resource constraints, such as inadequate funding, outdated infrastructure, and shortages of technical expertise[1–3]. Additionally, AI algorithms typically rely on large and high-quality datasets for training and validation. However, LMIC hospitals may have limited access to comprehensive and digitized healthcare

[1]Department Engineering Science, Institute of Biomedical Engineering, University of Oxford, Oxford, UK. [2]Hospital for Tropical Diseases, Ho Chi Minh, Vietnam. [3]National Hospital for Tropical Diseases, Hanoi, Vietnam. [4]Oxford University Clinical Research Unit, Ho Chi Minh, Vietnam. [5]Oxford Cancer & Haematology Centre, Oxford University Hospitals NHS Foundation Trust, Oxford, UK. [6]Department of Oncology, University of Oxford, Oxford, UK. [7]Centre for Tropical Medicine and Global Health, University of Oxford, Oxford, UK. [8]Oxford-Suzhou Centre for Advanced Research (OSCAR), Suzhou, China. [9]These authors contributed equally: Louise Thwaites, David A. Clifton. ✉e-mail: jenny.yang@eng.ox.ac.uk

data[2–5]. These resource limitations pose significant challenges for the adoption and implementation of healthcare AI systems, especially when compared to many HIC hospitals. As such, many algorithms trained on data outside of an LMIC context (such as those trained using HIC data) are being applied to LMIC settings[3,6]. However, without adequate consideration of the unique contexts and requirements of LMICs, these systems may struggle to achieve generalizability and widespread effectiveness[3,5–7].

Machine learning (ML) generalization refers to a model's ability to accurately apply its learned knowledge from training data to new, unseen data[8]. This capability is particularly valuable when models are deployed in real-world scenarios, where they must perform well on independent datasets encountered in real-time. In clinical contexts, two common types of generalizability are temporal generalizability (applying prospectively within the center where a model was developed) and external/geographic generalizability (applying a model at an independent center). In this study, we will focus on external/geographic generalizability.

While achieving broad generalizability is desirable for scalability, cost-effectiveness, and applicability to diverse cohorts/environments, it is often not feasible. Achieving external generalizability is challenging due to population variability (patients at one center may not represent those in another location)[7,9,10], healthcare disparities (variations in access to healthcare services, quality of care, and healthcare infrastructure)[6,11,12], variations in clinical practice (local guidelines, healthcare systems, and cultural factors)[8–10], and differences in data availability and interoperability (limited access to comprehensive and standardized data, variations in data formats, coding systems, and collection processes)[1,3–5,12]. These differences are especially apparent when comparing HIC and LMIC hospitals.

In order to achieve optimal integration and effectiveness of AI development in LMICs, it is imperative to adopt tailored approaches and strategies that specifically address the unique contexts of LMICs[3–5,12]. With a particular emphasis on biomedical engineering and AI, we aim to evaluate the feasibility of generalizability, specifically when deploying a model that was initially developed in a HIC setting to an LMIC setting. Our goal is to explore practical solutions that demonstrate effective performance while also investigating the ways in which international collaborations can offer optimal support for these development initiatives.

The collaboration between the Oxford University Clinical Research Unit (OUCRU) in Ho Chi Minh City, Vietnam, The University of Oxford Institue of Biomedical Engineering in Oxford, England, the Hospital for Tropical Diseases in Ho Chi Minh, Vietnam, and the National Hospital for Tropical Diseases in Hanoi, Vietnam, aims to improve the provision of critical care in LMIC settings. Their primary objective is to accurately identify patients requiring critical care and enhance the quality of care they receive, thereby addressing the unique challenges encountered within LMIC healthcare systems. Thus, in this study, we evaluate the performance of a United Kingdom (UK)-based AI system on patients in Vietnam.

Previously, we developed an AI-driven rapid COVID-19 triaging tool using data across four United Kingdom (UK) National Health Service (NHS) Trusts[8–10,13–15]. As such, through our collaboration with Vietnam-based centers, we aimed to translate the UK-based models to LMIC settings, specifically at the Hospital for Tropical Diseases (HTD) in Ho Chi Minh, Vietnam, and the National Hospital for Tropical Diseases (NHTD) in Hanoi, Vietnam.

In the UK, the NHS utilized a green-amber-blue categorization system, where green indicated patients with no COVID-19 symptoms, amber indicated patients with potential COVID-19 symptoms, and blue indicated laboratory-confirmed COVID-19 cases. Through a validation study conducted at the John Radcliffe Hospital in Oxford, England, we demonstrated that our AI screening model improved the sensitivity of lateral flow device (LFD) testing by ~30%, and correctly excluded 58.5%

of negative patients who were initially triaged as COVID-19-suspected by clinicians[14]. Furthermore, the AI model provided diagnoses, on average (median), 16 min (26.3%) earlier than LFDs, and 6 h and 52 min (90.2%) earlier than Polymerase Chain Assay (PCR) testing, when the model predictors were collected using point of care full blood count analysis. Applying a similar screening tool at the HTD and NHTD in Vietnam could offer a systematic approach to prioritize and manage patient care. It would allow for the efficient use of limited resources, including clinician expertise, ventilators, and beds, ultimately optimizing patient outcomes and ensuring timely access to appropriate interventions. These benefits are especially valuable in LMIC settings where resource constraints pose significant challenges to healthcare delivery[16].

Furthermore, building upon the four UK datasets, we have conducted prior research exploring the generalizability of models across different hospital sites[8]. Specifically, we investigated how well pre-existing models developed in one hospital setting performed when applied to another location. To accomplish this, we introduced three distinct methods: (1) utilizing the pre-existing model without modifications, (2) adjusting the decision threshold based on site-specific data, and (3) fine-tuning the model using site-specific data through transfer learning. Our findings revealed that transfer learning yielded the most favorable outcomes, indicating that customizing the model to each specific site enhances predictive performance compared to other pre-existing approaches.

Through this COVID-19 case study, we now evaluate the feasibility of adapting models in hospitals that span diverse socioeconomic brackets, additionally evaluating corresponding datasets obtained from two hospitals in Vietnam. In doing so, we aim to expand the understanding of ML-based methods in identifying COVID-19 cases across different healthcare settings, thus contributing to the advancement of diagnostic capabilities in diverse regions. We particularly focus on transitioning a model from a HIC setting to a LMIC setting. By leveraging datasets sourced from four UK NHS Trusts and two hospitals located in Vietnam, we illustrate practical methodologies that can enhance the performance of models. Additionally, we highlight the importance of collaborative efforts in the development of resilient AI tools tailored to healthcare systems in LMICs.

## Results

COVID-19 prevalences observed at all four UK sites during the data extraction period ranged from 4.27% to 12.2%. COVID-19 prevalence was highest in the BH cohort, owing to the evaluation timeline spanning the second UK pandemic wave during January 1, 2021 to March 31, 2021 (12.2% vs 5.29% in PUH and 4.27% in UHB; Fisher's exact test $p < 0.0001$ for both). Prevalance at the Vietnam sites was significantly higher (74.7% and 65.4% at HTD and NHTD, respectively, $p < 0.0001$), as these were exclusively infectious disease hospitals, and handling the most severe cases of COVID-19.

Between all UK and Vietnam cohorts, all matched features had a significant difference in population median (Kruskal-Wallis, $p < 0.0001$). In the case of features exclusive to the UK cohorts, a significant distinction in population median was observed for all features, except for mean cell volume, where the population median appeared to be similar ($p = 0.210$). Full summary statistics (including median and interquartile ranges) of vital signs and blood tests for all patient cohorts are presented in Supplementary Tables 2 and 3, respectively.

It is important to highlight that, upon a preliminary examination of the summary statistics of the datasets, we observed the presence of extreme values in the Vietnam datasets. For instance, the minimum hemoglobin value was recorded as 11 g/L, which is notably rare, as values this low are typically considered highly unlikely[17–19]. Another instance is observed in the white blood cell count feature, where the dataset's maximum value was registered at 300, an exceptionally

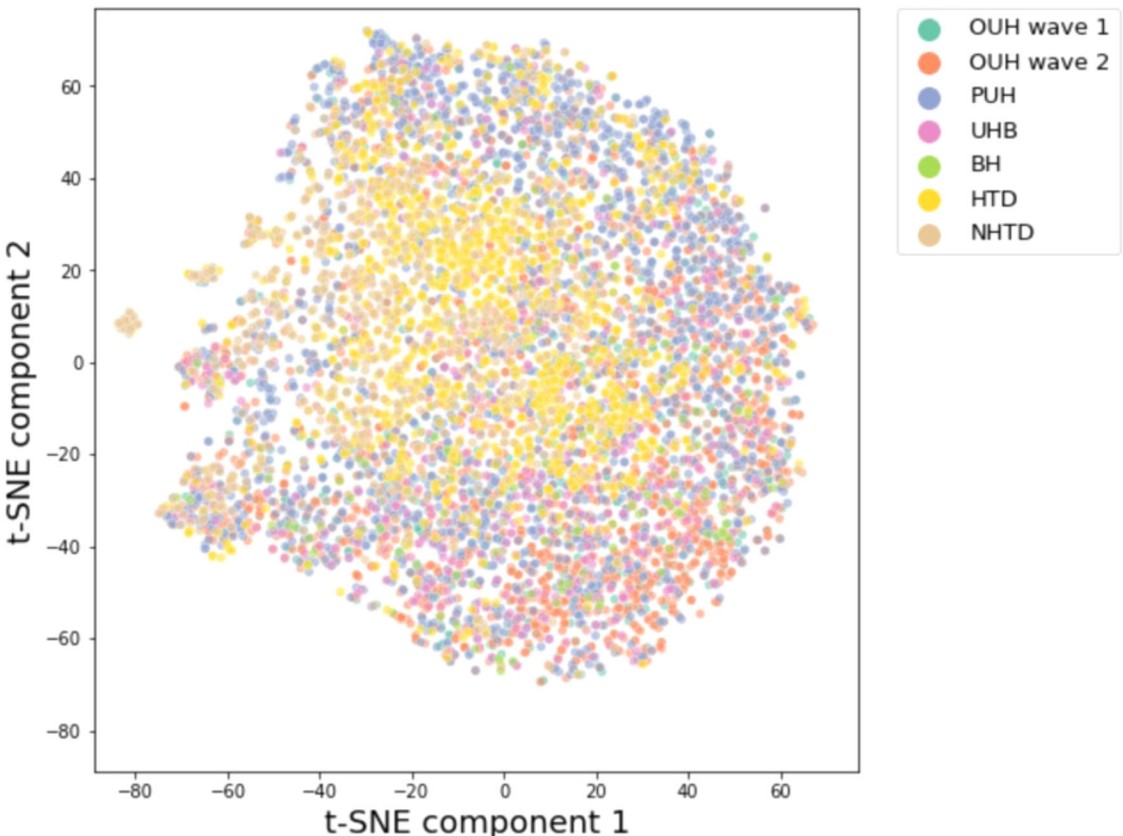

**Fig. 1 | t-SNE plot of UK and Vietnam datasets with reduced feature set.** Plot includes all positive COVID-19 samples in UK and Vietnam datasets, including the matched/reduced set of features.

extreme value[19]. While such levels of deviation theoretically can occur in cases of hematological malignancy, they remain exceedingly rare occurrences. In the Vietnam datasets, there were some extreme values in patients with lymphoma. For our experiments, we made a deliberate choice to retain these extreme values in the dataset. This decision was motivated by our aim to evaluate the performance of models using real-world data, acknowledging the presence of extreme values and potential errors (this is further discussed in "Discussion").

**Reduced feature set**

We initially employed t-Stochastic Neighbor Embedding (t-SNE) to generate a low-dimensional representation of all positive COVID-19 cases within each hospital cohort. As depicted in Fig. 1, there are no immediately discernible indications of site-specific biases or distributions apparent in the visualization, as evidenced by the absence of distinct clusters.

Following the training of models on the OUH pre-pandemic and wave one data, we conducted prospective and/or external validation on six datasets. As anticipated, when utilizing the reduced dataset based on the available features in Registry (database for hospitals in Vietnam, further described in "Methods"), the performance of the models was ~5-10% lower in terms of AUROC compared to previous studies using the same training and test cohorts. The AUROC ranges were as follows: OUH (0.784–0.803), PUH (0.812–0.817), UHB (0.757–0.776), BH (0.773–0.804), in contrast to the results reported in prior research[8,9,13]: OUH (0.866–0.878), PUH (0.857–0.872), UHB (0.858–0.878), BH (0.880–0.894). The AUROC scores remained relatively consistent across all UK test sets, with a standard deviation (SD) of 0.017 for the NN model. However, the AUROC was lower for the HTD and NHTD centers, with an NN AUROC of 0.577 (CI 0.551–0.604) and 0.515 (0.491–0.541), respectively.

Although we optimized the classification threshold for a sensitivity of 0.85, sensitivity scores varied across all test sets, with an SD of 0.090 for the NN model. The highest sensitivities were observed at HTD, PUH, and NHTD (0.908, 0.835, 0.831 for the NN model, respectively), while the lowest sensitivities were observed at OUH, UHB, and BH (0.718, 0.690, 0.688 for the NN model, respectively). Even within the same country, there was a significant range in sensitivity, with ranges of 0.688–0.835 for UK centers and 0.831-0.908 for Vietnam centers in the NN model. In the UK test sets, specificity exhibited a reasonable balance with sensitivity. However, for the Vietnam datasets, specificity was notably poor, with values of 0.139 (0.114–0.167) and 0.159 (0.134–0.185) for NN models at HTD and NHTD, respectively.

Consistent with previous studies, our models achieved high prevalence-dependent negative predictive value (NPV) scores (>0.944) on the UK datasets, demonstrating their ability to confidently exclude COVID-19 cases.

We conducted a subgroup analysis for both correct and incorrect classifications across COVID-19-negative and COVID-19-positive groups, focusing on the features used for prediction. This was evaluated on the neural network model. Patients with lower white blood cell counts exhibited higher false negative rates at both HTD and NHTD. At NHTD, hemoglobin and platelet values also showed notable differences in distribution regarding false positive and false negative rates. Detailed prediction distribution plots by class are available in Section D of the Supplementary Material.

We conducted an additional sensitivity analysis to address the uncertainty surrounding the viral status of patients who underwent rapid antigen testing or where the testing method was unspecified at NHTD. Utilizing the NN model, which demonstrated superior performance, and evaluating solely on the subset of NHTD patients with confirmed PCR testing, we attained AUROC scores of 0.492

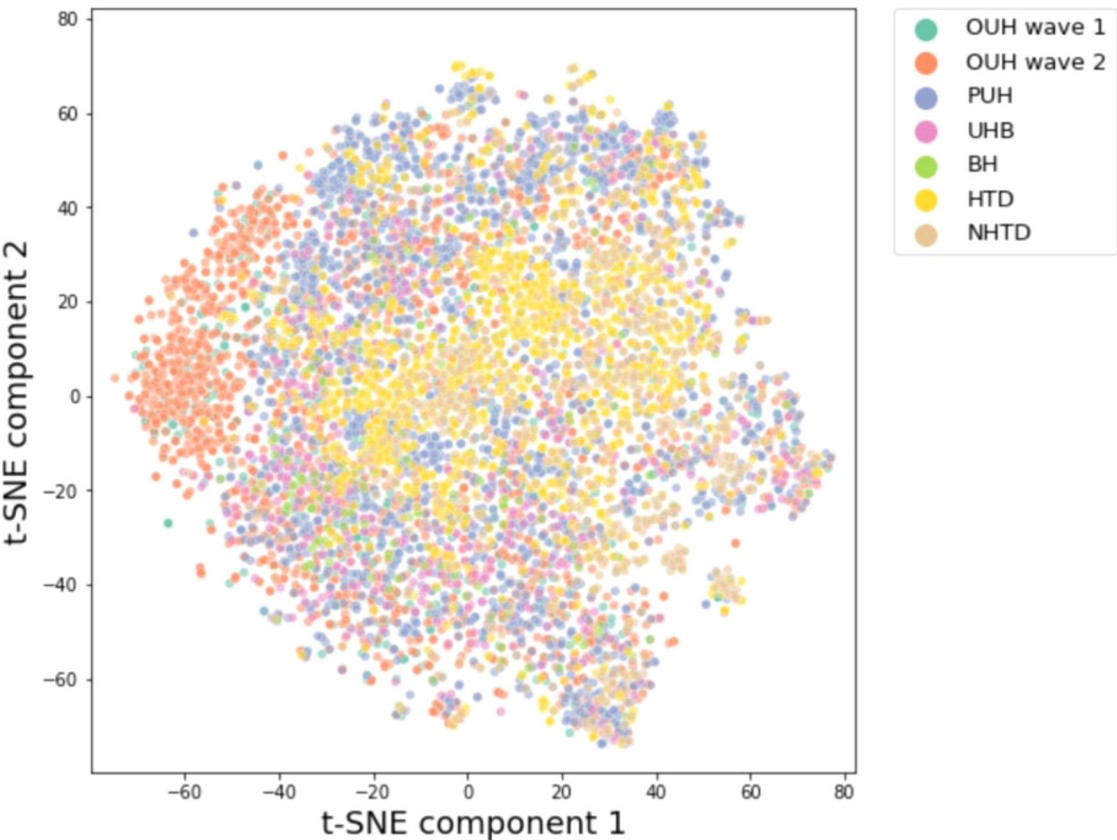

**Fig. 2 | t-SNE plot of UK and Vietnam datasets with comprehensive feature set.** Plot includes all positive COVID-19 samples in UK and Vietnam datasets, including the comprehensive set of features, which were generated using the GATS technique.

(0.445–0.548) for the NHTD set. Generally, we observed comparable results, indicated by overlapping confidence intervals, when compared to datasets incorporating alternative testing methods. Consequently, further experiments were conducted using the dataset encompassing testing methods beyond PCR.

## Comprehensive feature set

Upon the inclusion of additional UK features, which were generated using GATS (further described in "Methods") for the Vietnam datasets, it becomes evident that a separate cluster emerges during t-SNE visualization, represented by the orange data points corresponding to the OUH wave two cohort in Fig. 2. This observation implies that the training data can be grouped together based on, and consequently exhibits bias towards, site-specific features. These features could encompass factors such as annotation methods, data truncation techniques, the type of measuring devices utilized, or variances in data collection and processing tools. It is worth noting that a similar observation was also made in a prior studies that employed different stratifications of the same datasets[8–10].

Upon utilizing the comprehensive set of features, including the filling in of missing Registry values using $k$ nearest neighbors (kNN) and GATS, our models exhibited improvements of up to 10% on the UK test sets ($p < 0.001$), as shown in Fig. 3. These improvements resulted in achieving comparable AUROC scores to those reported in previous studies that employed the same training and test cohorts. The ranges of AUROC scores were as follows: OUH (0.854–0.877), PUH (0.832–0.877), UHB (0.846–0.860), BH (0.875–0.905), compared to the results reported in previous studies[8,9,13]: OUH (0.866–0.878), PUH (0.857–0.872), UHB (0.858–0.878), BH (0.880–0.894). The AUROC scores remained relatively consistent across all UK test sets, with the XGB model

exhibiting the best performance, with a standard deviation (SD) of 0.019. Similar to previous findings, the AUROC scores were lower at the HTD and NHTD centers. Nonetheless, the NN model outperformed the XGB and LR models by an approximate margin of 5%, achieving AUROC scores of 0.590 (0.563–0.617) for HTD and 0.522 (0.497–0.544) for NHTD, respectively. This represented an improvement from the scores of 0.577 (0.551–0.604) ($p = 0.033$) for HTD and 0.515 (0.491–0.541) ($p = 0.409$) for NHTD when using the reduced datasets. The AUPRC also demonstrated improvement across all test sites, with a notable improvement of over 25% at BH.

In terms of sensitivity, the scores were more consistent across the UK datasets, with a range of 0.779–0.825 and an SD of 0.021 for the XGB model. Across the Vietnam datasets, sensitivity ranged from 0.610–0.646 for the XGB model and 0.660–0.661 for the NN model, indicating increased consistency compared to previous experiments.

For the UK test sets, specificity remained reasonably balanced with sensitivity. In the case of the Vietnam sites, specificity improved and became slightly more balanced with sensitivity, with values of 0.465 (0.426–0.505) and 0.353 (0.319–0.385) for the NN model at HTD and NHTD, respectively. However, this improvement corresponded to a decrease in sensitivity.

Consistent with previous studies, our models achieved high prevalence-dependent NPV scores (>0.951) on the UK datasets, affirming their capability to confidently exclude COVID-19 cases.

## Transfer learning

When we applied transfer learning to adapt models developed in the UK to the local context of Vietnam, we observed improved classification performance at both centers. This improvement was evident in both the prospective validation on the center used for transfer learning and the external validation on the other center.

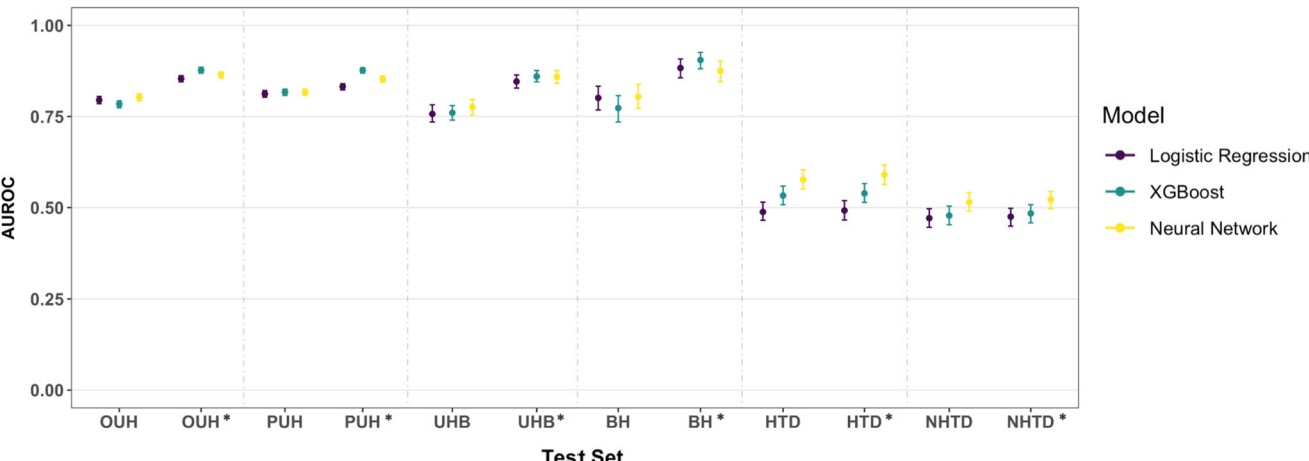

**Fig. 3 | COVID-19 diagnosis performance across logistic regression, XGBoost, and neural network models trained on the UK data.** Results are presented as AUROC for the reduced feature set and the comprehensive feature set (GATS-filled), with * representing the comprehensive dataset. Error bars are shown as 95% confidence intervals (CIs), which are computed using 1000 bootstrapped samples drawn from each test set. Source data are provided as a Source Data file.

When using the reduced feature set for training, we found that AUROC improved from 0.577 (0.551–0.604) to 0.707 (0.654–0.756) for HTD ($p = 0.001$) and from 0.515 (0.491–0.541) to 0.653 (0.627–0.677) for NHTD ($p < 0.001$), when pre-trained on a subset of the HTD data. Pre-training models using a subset of the NHTD data also yielded improvements, albeit slightly lower, achieving AUROCs of 0.656 (0.599–0.712) ($p < 0.001$) for HTD and 0.650 (0.623-0.675) for NHTD ($p < 0.001$).

AUPRC scores also showed improvements across all centers, with particularly notable improvements of 7–15% at NHTD. In terms of sensitivity, we observed improved performance with less variation across the two hospitals, with a difference of less than 2%.

Sensitivity significantly improved compared to applying ready-made models without transfer learning (improved between 0.10 and 0.20 across both sites, $p < 0.001$). Specificity did not exhibit any improvement at HTD (range 0.386–0.418 compared to 0.465 (0.426–0.505) without transfer learning), however appeared to show slight improvement at NHTD (range 0.328–0.422 compared to 0.353 (0.319–0.385) without transfer learning).

When we repeated the transfer learning experiments using the comprehensive feature set, including the filling in of missing Registry values using kNN and GATS, we observed further improvements of 1–3% in both AUROC and AUPRC across all iterations (0.113 < $p$ < 0.180, compared to models trained without GATS). However, there was no clear pattern in the improvement of sensitivity and specificity.

In order to assess the value of transfer learning, we conducted a comparison with the alternative approach of developing a model locally in Vietnam, starting from scratch and using only the available data from within the country.

When training a model locally at HTD, using the features available in Registry, we observed improvements in AUROC compared to using a UK-based model trained at OUH. The AUROC improved from 0.577 (0.551–0.604) to 0.664 (0.613–0.716) during prospective validation at HTD ($p = 0.032$), and from 0.515 (0.491–0.541) to 0.639 (0.615–0.663) during external validation at NHTD ($p < 0.001$). Similarly, when trained locally at NHTD, the AUROC improved to 0.608 (0.585–0.634) during external validation at HTD ($p < 0.001$) and 0.662 (0.604–0.717) during prospective validation at NHTD ($p < 0.001$). AUPRC also showed improvements, particularly at NHTD, with enhancements of up to 16%. These improvements are shown in Figs. 4 and 5.

In terms of sensitivity, there was improved performance with less variation across the two hospitals, with a difference of less than 2%.

Models trained at HTD exhibited higher sensitivity (ranging from 0.849 to 0.868) compared to those trained at NHTD (ranging from 0.760 to 0.786), but this was accompanied by a trade-off in specificity, with the model trained at NHTD demonstrating superior specificity (ranging from 0.378 to 0.455) compared to the model trained at HTD (ranging from 0.305 to 0.369).

When compared to transfer learning, the locally-trained models (trained solely on the data available at the site) exhibited slightly lower performance. Using the same features for model development (reduced feature set available in HTD and NHTD hospital systems), the transfer learning model (finetuned at HTD) achieved an AUROC of 0.707 (0.654–0.756) when tested at HTD, while the HTD locally-trained model achieved an AUROC of 0.664 (0.613–0.716) ($p = 0.01$). When evaluating on NHTD data, the transfer learning model (finetuned at HTD) achieved an AUROC of 0.653 (0.627–0.677), while the HTD locally-trained model achieved an AUROC of 0.639 (0.615–0.663). Similarly, when models were trained locally or finetuned (via transfer learning) at NHTD, the transfer learning model achieved an AUROC of 0.656 (0.599–0.712) during HTD testing, whereas the NHTD locally-trained model achieved an AUROC of 0.608 (0.585–0.634). However, when testing on NHTD, both models achieved similar scores, with a slightly higher AUROC of 0.662 (0.604–0.717) for the NHTD locally-trained model compared to 0.650 (0.623–0.675) for the transfer learning model ($p = 0.458$).

Overall, the best performing models were those using transfer learning (especially with the comprehensive dataset), achieving an AUROC range of 0.663–0.727 across all iterations.

Although the subset of HTD and NHTD data used in testing varied slightly among different methods (either the complete dataset or 60% of the data was employed for testing), sensitivity analysis yielded AUROC scores of 0.577 (0.551–0.604) and 0.562 (0.509–0.616) for the full and partial (prospective) HTD data variations, respectively. Similarly, during the senstivity analysis of NHTD, AUROC scores of 0.515 (0.491–0.541) and 0.489 (0.428–0.549) were obtained for the full and partial (prospective) variations, respectively. While the utilization of full test sets seemed to enhance the apparent accuracy of the models, the comparable results, as indicated by overlapping confidence intervals, underscored the stability of the models across both the complete test sets and their respective subsets.

## Discussion

Using ready-made HIC models (UK models) in LMIC settings (Vietnam hospitals) without customization resulted in the lowest predictive

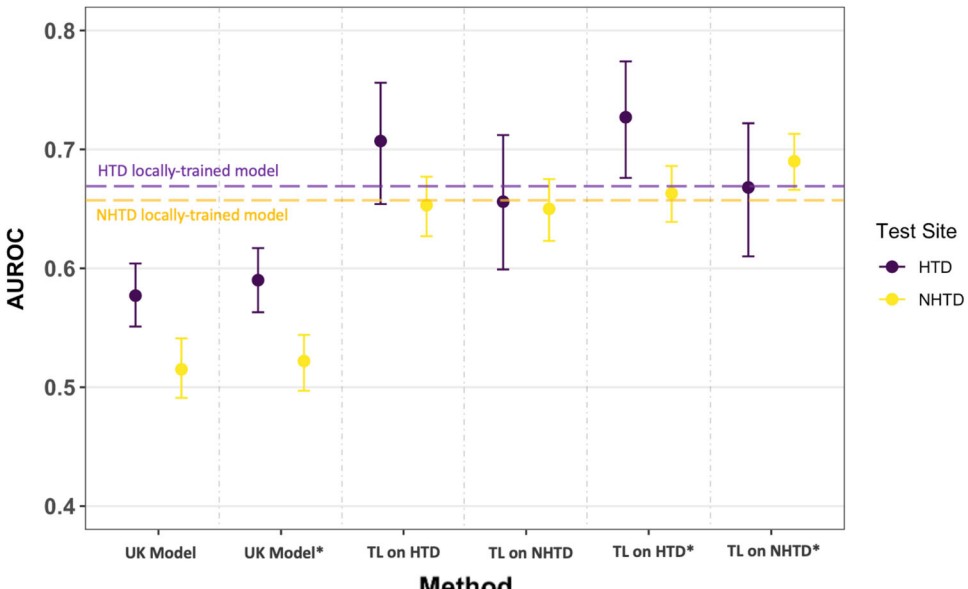

**Fig. 4 | COVID-19 diagnosis AUROC performance at HTD and NHTD using neural network models which were ready-made (the UK-based models) and models which were fine-tuned using transfer learning.** Models trained and tested locally at HTD and NHTD are represented by the horizontal purple and yellow dotted lines, respectively. Results are presented for the reduced feature set and the comprehensive feature set (GATS-filled), with * representing the comprehensive dataset. Error bars are shown as 95% confidence intervals (CIs), which are computed using 1000 bootstrapped samples drawn from each test set. Source data are provided as a Source Data file. TL transfer learning.

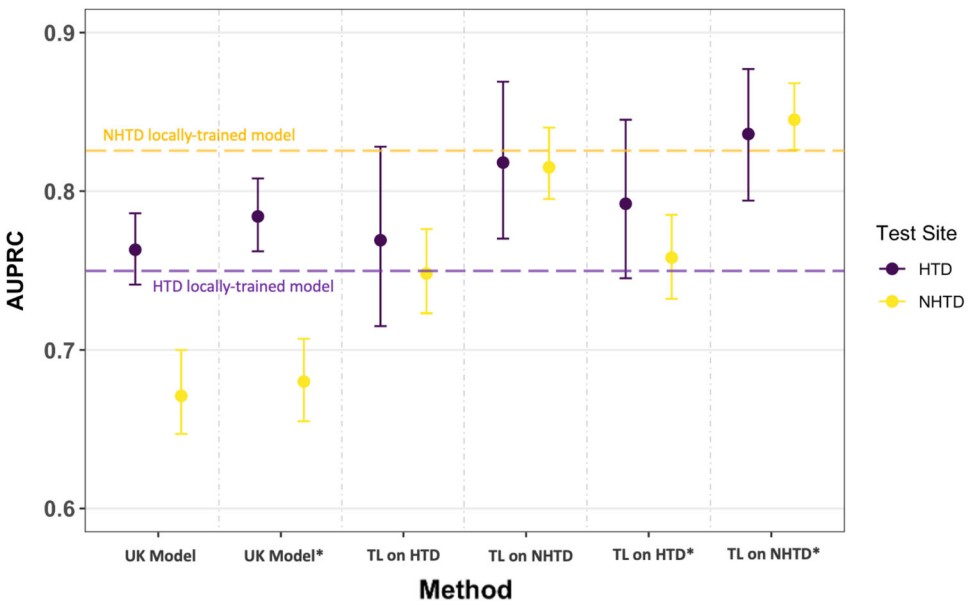

**Fig. 5 | COVID-19 diagnosis AUPRC performance at HTD and NHTD using neural network models which were ready-made (the UK-based models) and models which were fine-tuned using transfer learning.** Models trained and tested locally at HTD and NHTD are represented by the horizontal purple and yellow dotted lines, respectively. Results are presented for the reduced feature set and the comprehensive feature set (GATS-filled), with * representing the comprehensive dataset. Error bars are shown as 95% confidence intervals (CIs), which are computed using 1000 bootstrapped samples drawn from each test set. Source data are provided as a Source Data file. *TL* Transfer Learning.

performance and the highest variability in AUROC/AUPRC and sensitivity/specificity. This finding aligns with a previous study[8] that focused on external validation of COVID-19 prediction models within the UK. Additional research has similarly indicated that model performance declined when models trained on data from contexts different from the implementation setting were employed, including transitions from HIC to LMIC settings[3,6]. Thus, these outcomes were anticipated, as diverse hospital settings can significantly differ in terms of unobserved factors, protocols, and cohort distributions, posing challenges to model generalization. Despite potential similarities in human pathophysiology for specific outcomes, neural networks heavily rely on the specific datasets and patient cohorts used during training[8–10]. Therefore, considering the unique attributes of each setting is crucial for achieving optimal model performance. In particular, the datasets analyzed in this study exhibited variations in patient demographics, genotypic/phenotypic characteristics, and other determinants of

health, such as environmental, social, and cultural factors. For example, the HTD and NHTD datasets were primarily composed of Southeast Asian (Vietnamese) patients, which may have influenced the models' generalization capabilities (as opposed to the UK datasets, which had a majority of patients from a white demographic). Furthermore, as we conducted error analysis on subgroups related to classification features, it is imperative for future research to extend this analysis to demographic subgroups. This would enhance our comprehension of how various groups might experience differential impacts from a machine learning algorithm (we did not have these features fully available in the datasets, and thus were not able to perform these analyses).

In our research, we consistently observed superior performance of the neural network model when applied to the Vietnam datasets. Nonetheless, it's important to recognize the tendency of neural networks to overfit. Despite employing a straightforward network architecture with only one hidden layer, the risk of overfitting increases when training data is limited, potentially resulting in poor generalization. Hence, it remains crucial to assess simpler models like Logistic Regression (LR) and XGBoost as benchmarks, as demonstrated in our analysis.

We found that transfer learning performed the best in terms of COVID-19 diagnosis and generalizability across both the UK and Vietnam hospital sites. This method becomes particularly valuable for LMIC hospitals that often encounter difficulties in gathering an adequate amount of data or resources to train machine learning models effectively[20]. By leveraging transfer learning, LMIC hospitals can harness collaborative efforts with HIC centers, benefiting from their expertise and resources while adapting the models to local contexts with limited data availability. This approach allows for the development of tailored models using smaller datasets, addressing the challenges faced by LMIC hospitals.

It is important to highlight that the development of site-specific models (models trained on data from the local context) also yielded strong performance, ranking as the second-best approach. In the case of HTD and NHTD, when subjected to prospective validation at the site where the model was originally developed, the models demonstrated superior performance compared to external validation conducted at a different site. This observation aligns with expectations since datasets from external sites can possess distinct underlying data distributions and statistical characteristics, influencing the generalizability and performance of the models[5,6,8]. While models trained at a central location (such as a HIC like the UK), may offer certain advantages like data availability, efficiency, and scalability, there is significant merit in the development of AI models that are finely tuned to the intricacies of their particular operational environment and the specific context of their deployment. This is particularly critical when considering LMIC settings, as AI models trained exclusively on HIC data may introduce biases into AI outputs, potentially resulting in subpar performance[5], which we've demonstrated in our experiments.

When using GATS, we found that models exhibited further improvements at HTD and NHTD during transfer learning and external validation using the UK-based models. Therefore, data generation methods, such as GATS, provide promising solutions for tackling missing data challenges in LMIC hospitals. Utilizing this technique enables the generation of complete datasets, which in turn facilitates effective model training. The selection of features to be added can be guided by those that have proven to be effective in models developed in HIC settings. However, it is important to acknowledge that despite using kNN to match patients based on similar features, some bias still persists as missing values are being filled using UK datasets, which have their own distinct distributions (recall the t-SNE representation, where the UK features were clustered together). This may explain why even though GATS slightly improved apparent accuracy during transfer learning, results were

not found to be significant between transfer learning with and without GATS. However, results obtained when evaluating UK models (without any transfer learning step) were found to have significant improvement with the addition of GATS. Hence, careful consideration and scrutiny are necessary to account for any potential bias introduced during the data generation process.

Furthermore, it is important to recognize that while HICs typically possess extensive collections of health data, many LMICs face limitations in data availability, particularly regarding the volume and quality of data accessible electronically and the asynchronous, varied nature of information. These factors can make it challenging to train AI models[5,21]. Therefore, in the context of LMICs, where datasets may be smaller and data accessibility issues persist, it is advisable to consider additional computational techniques such as GATS to better leverage and optimize the utilization of available data resources, and ultimately improve the effectiveness and generalizability of HIC-trained models in LMIC settings.

Regarding data quality, we also detected the presence of outliers within the Vietnam datasets, such as the minimum recorded hemoglobin value of 11 g/L. This particular value would typically be considered highly improbable[17,18]. The existence of such outliers could be attributed to a unit conversion error, where values were erroneously shifted by a factor of 10 (some locations utilize g/dL instead of g/L), or they may be the result of data entry errors. Since we aimed to work with real data, our model incorporates such instances of incorrect data entry and outliers. In the case of extreme values for white blood cell count, there were some extreme values found in patients with lymphoma in the HTD and NHTD datasets. In certain scenarios, outliers like these may contain unique information that can enhance a model's ability to generalize effectively, rendering the models more robust and less susceptible to noise. The decision of whether to retain extreme values in a dataset or not depends on the context and the problem under consideration. Extreme values can indeed offer valuable information, but it is important to handle them appropriately to prevent any adverse impact on model performance[22,23]. Therefore, for future studies, it may be worthwhile to explore additional filtering and preprocessing steps to address these anomalies and enhance the dataset's quality before model development and testing.

It is essential to consider that HTD and NHTD are specialized hospitals for infectious diseases. They specifically designated as COVID-19 hospitals during the pandemic, primarily receiving referrals for severe cases of COVID-19. While both the UK and Vietnam datasets included the first recorded blood tests and observations, it is important to acknowledge that in LMICs during pandemics, there might be some delay in recording these features after the initial presentation. Moreover, COVID-19 negative cases in these facilities typically involved other infectious diseases, and critical cases, including patients with various comorbidities, were treated at these hospitals. Given that the Vietnamese cohorts primarily consisted of severely ill patients, this might account for the more noticeable fluctuations in blood test results. Due to these differences, models may encounter challenges in accurately differentiating COVID-19 for patients at HTD and NHTD based on vital signs and blood test features, as other diseases (including infectious diseases) might also be present. Furthermore, in the case of UK hospitals, there was a broader spectrum of COVID-19 case severity. The UK datasets encompassed all individuals coming to the hospital, with only a small subset of patients progressing to ICUs. Consequently, diagnosing COVID-19 using AI is a significantly more challenging task at HTD and NHTD because we must distinguish the specific reason for ICU admission, particularly in cases of infectious diseases. For instance, distinguishing COVID-19 from bacterial pneumonia (which is frequently encountered at HTD and NHTD) is more challenging than distinguishing it from a case like a fractured leg.

This difficulty may also account for the lower level of specificity observed in the HTD and NHTD datasets compared to the UK sites.

Thus, even if AUROC/AUPRC metrics are high at external sites, it may be necessary to tailor the classification threshold (i.e., the criterion for categorizing COVID-19 status as positive or negative) for each site independently, to maintain the desired levels of sensitivity and specificity[8]. Nonetheless, we acknowledge the value of assessing the likelihood of having a disease rather than simplifying it into a binary classification. While we opted for a binary classification to expedite the categorization of COVID-19 as positive or negative, probability can serve as a viable final outcome for tasks when suitable. This is particularly relevant given that the Vietnam datasets contained information on varying levels of disease severity. Future studies can consider harnessing these labels to offer more detailed diagnoses or to estimate levels of uncertainty when necessary.

While we analyzed patient cohorts admitted to ICUs at HTD and NHTD, the datasets and features we utilized were those readily available and documented upon hospital admission. These models can provide swift insights and facilitate efficient and precise triage during a patient's initial presentation at the hospital. It's important to note that in many cases, such as those observed in Vietnam, by the time patients are transferred from the hospital to the ICU, the diagnosis is typically already established. Therefore, even though similar features are recorded upon ICU admission, in these scenarios, the relevance of a machine learning-based classification algorithm may appear redundant, and the benefits of diagnosing at ICU admission may be limited. Ultimately, the decision to employ machine learning algorithms should consider various factors, including the clinical context, the patient's condition, and the urgency of the situation. Additionally, similar approaches could be applied to other diseases or integrated into local hospital protocols, including guidelines for patient transfer, among other considerations.

It is also important to acknowledge that prediction models can never be fully validated due to inherent variability in their performance across different locations, settings, and time periods[20,24]. A single external validation study conducted in a specific geographical area, during a particular time frame, and within a distinct patient population offers only a limited view and cannot assert universal applicability. In this study, our investigation spanned a significant time period, from December 1, 2019, to December 30, 2022. During this extended duration and particularly during peak pandemic periods, such as the COVID-19 outbreak, the relationship between patient and disease factors with clinical events, including hospital-acquired infections, may undergo changes[20]. Additionally, over time, there may be variations in practice patterns such as hardware and software updates and changes in protocols, which can impact data capture and outcomes. Although this retrospective study offered valuable insights into historical data, future research should ideally focus on prospective analysis. Models should be updated regularly to maintain their relevance. This approach enables a more dynamic assessment of model performance and provides timely feedback for refining and improving predictive models. Therefore, future efforts should validation efforts should aim to quantify and comprehend the heterogeneity in model performance, rather than solely focusing on point estimates[24]. This broader understanding of performance variability is crucial for refining and improving the models over time. For instance, in LMIC settings, real-time data preprocessing and curation can be achieved through cost-effective and accessible strategies. In the study highlighted here, an offline, in-house version of the algorithm can be used, where a doctor manually enters feature values in real-time (feasible with only 14 features). These values can then be automatically processed through a script that imputes missing features and performs standardization, ultimately outputting a diagnosis for further triaging. Additionally, emphasizing the use of open-source tools and scalable, cost-effective infrastructure ensures applicability in resource-constrained settings.

Finally, the adoption of AI in LMICs encounters significant infrastructural and capacity-building challenges[1,2,4,5]. These challenges encompass power outages, unreliable internet connectivity, cybersecurity concerns, inadequate digital infrastructure (such as data and storage), and a shortage of skilled AI professionals. As a result, prioritizing AI solutions may divert resources from more urgent foundational needs. These issues also impact the broader concern of AI governance, which remains a challenge even in HICs[25], and is likely even more challenging in LMICs. Therefore, while AI holds promise, its adoption in LMICs necessitates a careful, context-sensitive approach to address these underlying challenges.

## Methods

In this study, we used clinical data with linked, deidentified demographic information for patients across hospital centres in the UK and Vietnam. United Kingdom National Health Service (NHS) approval via the national oversight/regulatory body, the Health Research Authority (HRA), has been granted for use of routinely collected clinical data to develop and validate artificial intelligence models to detect Covid-19 (CURIAL; NHS HRA IRAS ID: 281832). The study was limited to working with deidentified data, and extracted retrospectively; thus, explicit patient consent for use of the data was deemed to not be required, and is covered within the HRA approval. All necessary consent has been obtained and the appropriate institutional forms have been archived.

The ethics committees of the Hospital for Tropical Diseases (HTD) and the National Hospital for Tropical Diseases (NHTD) approved use of the HTD and NHTD datasets for COVID-19 diagnosis, respectively. The study was limited to working with deidentified data and collected as part of ongoing audit; thus, OxTREC (Oxford Tropical Research Ethics Committee), NHTD and HTD ethics committees have waived the need for individual informed consent for this process. All methods were carried out in accordance with relevant guidelines and regulations.

### Datasets

From the UK, we used data from hospital emergency departments in Oxford University Hospitals NHS Foundation Trust (OUH), University Hospitals Birmingham NHS Trust (UHB), Bedfordshire Hospitals NHS Foundations Trust (BH), and Portsmouth Hospitals University NHS Trust (PUH). For these datasets, United Kingdom National Health Service (NHS) approval via the national oversight/regulatory body, the Health Research Authority (HRA), has been granted for development and validation of artificial intelligence models to detect COVID-19 (CURIAL; NHS HRA IRAS ID: 281832). From Vietnam, we used data from the intensive care units (ICUs) in the Hospital for Tropical Diseases (HTD) and the National Hospital for Tropical Diseases (NHTD). This was approved by ethics committees of the HTD and the NHTD, respectively.

To ensure consistency with previous studies, we trained our models using the same cohorts as those used in[8–10,13,14]. Specifically, we utilized patient presentations exclusively from OUH for training and validation sets. Two data extracts were obtained from OUH, corresponding to the first wave of the COVID-19 epidemic in the UK (December 1, 2019, to June 30, 2020) and the second wave (October 1, 2020, to March 6, 2021) (Supplementary Section B). During the first wave, incomplete testing and the imperfect sensitivity of the PCR test resulted in uncertainty regarding the viral status of patients who were either untested or tested negative. To address this, similar to the approach taken in refs. 8–10,13,14, we matched each positive COVID-19 presentation in the training set with a set of negative controls based on age, using a ratio of 20 controls to 1 positive presentation. This approach created a simulated disease prevalence of 5%, which aligned with the actual COVID-19 prevalences observed at all four UK sites during the data extraction period (ranging from 4.27% to 12.2%). To account for the uncertainty in negative PCR results, sensitivity analysis

**Table 1 | Total patients and positive COVID-19 cases in the OUH training cohorts (OUH pre-pandemic and wave one), prospective validation cohort (OUH), external validation cohorts of patients admitted to three independent NHS Trusts (UHB, PUH, BH), and external validation cohorts of patients admitted to two Vietnam-based hospitals (NTD, NHTD)**

|  | Cohort | Total Patients | COVID-19 Positive Cases |
|---|---|---|---|
| OUH pre-pandemic | Before Dec 1/19 | 114,957 | 0 |
| OUH wave one | Dec 1/19-June 30/20 | 701 | 701 |
| OUH wave two | Oct 1/20-Mar 6/21 | 22,857 | 2012 (8.80%) |
| UHB | Dec 1/19-Oct 29/20 | 10,293 | 439 (4.27%) |
| PUH | Mar 1/20-Feb 28/21 | 37,896 | 2005 (5.29%) |
| BH | Jan 1/21-Mar 31/21 | 1177 | 144 (12.2%) |
| HTD | Dec 10/20-Dec 30/22 | 1820 | 1360 (74.7%) |
| NHTD | Nov 1/20-Dec 21/22 | 1611 | 1053 (65.4%) |

**Table 2 | Clinical predictors considered for COVID-19 diagnosis**

| Category | Matched UK and Vietnam | UK Features |
|---|---|---|
| Vital Signs | Heart rate, respiratory rate, systolic blood pressure, diastolic blood pressure, temperature | |
| Blood Test | Hemoglobin, hematocrit, white cell count, platelets | Mean cell volume, neutrophil count, lymphocyte count, monocyte count, eosinophil count, basophil count |
| Liver Function Tests & C-reactive protein | Bilirubin | Albumin, alkaline phosphatase, alanine aminotransferase, C-reactive protein |
| Urea & Electrolytes | Sodium, potassium, creatinine, urea | Estimated glomerular filtration rate |

was conducted and found to improve the apparent accuracy of the models, as described in refs. 10,14.

Thus, the model development process involved a dataset comprising 114,957 patient presentations from OUH prior to the global COVID-19 outbreak, guaranteeing that these cases are COVID-free. Additionally, we included 701 patient presentations that tested positive for COVID-19, as confirmed by a PCR test. This careful selection of data ensured the accuracy of COVID-19 status labels used during the training phase of the model.

We then validated the model on four UK cohorts (OUH wave two, UHB, PUH, BH), totaling 72,223 admitted patients (4600 COVID-19 positive), and two Vietnam cohorts (HTD and NHTD), totaling 3431 admitted patients (2413 COVID-19 positive). A summary of each respective cohort is in Table 1. Full inclusion and exclusion criteria are provided in the Supplementary Material.

For OUH, we included all patients presenting and admitted to the emergency department. For PUH, UHB, BH, HTD, and NHTD, we included all patients admitted to the emergency department. COVID-19 status at the UK sites and HTD was determined through confirmatory PCR testing, while at NHTD, both PCR and/or rapid antigen testing were used. Nonetheless, concerning NHTD, there were numerous instances where the specific test type was not recorded. Therefore, in order to maximize testing coverage, in cases where the test type was unspecified, we examined how COVID-19 was documented, including terms such as COVID-19 lower respiratory infection, COVID-19 pneumonia, SARS-COV-2 Infection, COVID-19 acute respiratory distress syndrome, Acute COVID-19, and others. For our analysis, alongside confirmatory testing, we considered any indication and severity of COVID-19 presence as COVID-19 positive. These diagnoses were confirmed by attending specialist infectious diseases clinicians, and thus, we consider these diagnostic labels to be robust. A detailed breakdown of the labels available within the NHTD database is provided in Supplementary Table 1. Furthermore, we conducted a sensitivity analysis for NHTD, comparing PCR-confirmed outcomes with those incorporating rapid antigen tests and other written documentation of COVID-19, which is detailed in "Results".

### Features

To facilitate a more meaningful comparison of our results with previous studies[8–10,14,15], we adopted a similar set of features. These features align with a focused subset of routinely collected clinical data, including the first recorded laboratory blood tests (comprising full blood counts, urea and electrolytes, liver function tests, and C-reactive protein) as well as vital signs.

Regarding the UK NHS datasets, it's worth noting that each hospital operates within its own distinct IT infrastructure. However, in general, laboratory data is managed within a system referred to as LIMS (Laboratory Information Management System). The data extraction process for these datasets typically involved sourcing data from either a LIMS mirror, a trust integration system that interfaces with LIMS, or a direct extraction from the LIMS system itself.

For the Vietnam hospitals, we extracted data from the Critical Care Asia Registry (we will refer to this as Registry, a dedicated prospectively acquired database facilitating quality improvement initiatives. To test model generalizability at HTD and NHTD, we had to match the features available at the UK hospitals to the features available in the NTD and NHTD system (i.e., those recorded on Registry).

Some features available in the UK datasets (such as albumin, alkaline phosphatase, C-reactive protein) are not routine tests on admission in HTD and NHTD.

Table 2 summarizes the final features included.

### Pre-processing

We first ensured uniformity in the measurement units for identical features. Next, we standardized all features to have a mean of 0 and a standard deviation of 1 to aid in achieving convergence in neural network models. To address missing values in the UK datasets, we used population median imputation. These steps are consistent with refs. 8–10,14,15. For matched features in the Vietnamese datasets, we also applied population median imputation. Additionally, we performed sensitivity analysis for these cohorts to account for missing values using the XGBoost model, which is the baseline model from previous studies and can handle missing values as input.

Using the test set, we achieved AUROC scores of 0.525 (0.500–0.553) and 0.427 (0.401–0.452) for the HTD and NHTD sets with missing values, respectively. For the imputed sets, the AUROC scores improved to 0.533 (0.508–0.559) and 0.478 (0.453–0.504) for HTD and NHTD, respectively. The technique used to handle completely missing features in the Vietnam datasets is discussed in the following sections.

## Model architectures

In order to evaluate the generalizability of developed models, we conducted investigations using three commonly used model architectures: logistic regression, XGBoost, and a standard neural network. Logistic regression is a linear model that is widely accepted in the clinical community; XGBoost is a tree-based model known for its strong performance on tabular data[26]; and lastly, a standard neural network serves as the foundation for many powerful machine learning models and can be used alongside transfer learning. It should be noted that LR is a relatively simple and linear classification model which does not inherently involve complex neural network architectures or deep learning, and thus, is not typically used alongside transfer learning; and XGBoost depends on the availability of the entire dataset, such that transfer learning is not typically feasible[8]. Additionally, a neural network has previously been shown to have superior performance for COVID-19 diagnosis (using the same UK cohorts)[8–10,15]. Thus, like previous studies, we trained a fully-connected neural network which used the rectified linear unit activation function in the hidden layers and the sigmoid activation function in the output layer. For updating model weights, the Adam optimizer was used during training. Details of the model architecture are presented in Section C of the Supplementary Material.

## Metrics

In order to evaluate the performance of the trained models, we provide the following metrics: sensitivity, specificity, positive predictive value, NPV, area under the receiver operating characteristic curve (AUROC), and area under the precision-recall curve (AUPRC). These metrics are accompanied by 95% confidence intervals (CIs), which are computed using 1000 bootstrapped samples drawn from the test set. Tests of significance ($p$ values) comparing model performances are calculated by evaluating how many times one model performs better than other models across 1000 pairs of bootstrapped iterations. We use 0.05 as the threshold for determining statistical significance.

We performed a grid search to adjust the sensitivity/specificity for identifying COVID-19 positive or negative cases. We chose to optimize the threshold to achieve sensitivities of 0.85 ($\pm$0.05), ensuring clinically acceptable performance in detecting positive COVID-19 cases. This chosen sensitivity surpasses the sensitivity of LFD tests, which achieved a sensitivity of 56.9% for OUH admissions between December 23, 2021, and March 6, 2021[14]. Additionally, the gold standard for diagnosing viral genome targets is real-time PCR, which has estimated sensitivities between 80 and 90%[27,28]. Thus, by optimizing the threshold to a sensitivity of 0.85, our models can effectively detect COVID-19 positive cases, comparable to the sensitivities of current diagnostic testing methods.

## Training outline

For each task, we utilized a training set to develop, select hyperparameters, train, and optimize the models. A separate validation set was employed for ongoing validation and threshold adjustment. Following successful development and training, six independent test sets were utilized to evaluate the performance of the final models.

To start, we used the OUH pre-pandemic controls and wave one positive cases to develop models, using the reduced feature set (i.e., matched HTD/NHTD features).

In their study, Soltan et al.[13] identified specific laboratory blood markers, such as eosinophils and basophils, as having a significant impact on model predictions. This determination was made through the application of SHAP (SHapley Additive exPlanations) analysis during the development and evaluation of their models using patient cohorts from the UK. However, these particular features were not accessible in the Registry dataset, and consequently, were not incorporated into the initial models developed for compatible testing across UK and Vietnam cohorts. We hypothesize that without the inclusion of these features during training, the models' performance would be inferior compared to the previously reported scores. Hence, our goal is to illustrate how HICs could potentially assist LMICs by facilitating dataset augmentation or completion, which can improve outcomes when applying an HIC model to an LMIC setting.

In the context of addressing missing data, nearest neighbor (NN) imputation algorithms provide efficient approaches for completing missing values. In these methods, each absent value in certain records gets replaced by a value derived from related cases within the entire dataset[29]. This approach has the capacity to substitute missing data with plausible values that closely approximate the true ones.

Drawing from a similar concept, a recent technique called Geometrically-Aggregated Training Samples (GATS)[30] has been introduced to address missing data challenges. GATS constructs training samples by blending various patient characteristics using convex combinations. This approach enables the creation of missing columns by combining features from multiple patient samples that do not have missing data in those columns. Importantly, these generated samples exist within the same data space as genuine training samples, preserving the original data structure and avoiding any distortion in the distribution of the imputed variables. This preservation facilitates effective model training, as these samples can be considered a summary of multiple patients. Furthermore, it's noteworthy that this method can be used to tackle missing columns without compromising the privacy of individual patient data, thereby mitigating privacy concerns.

We start by matching each patient in the HTD and NHTD datasets to the $k$ most similar patients in the UK datasets, based on the available features in the Registry. Here, we use the OUH wave two, PUH, UHB, and BH datasets, as to ensure that the training and test sets are completely independent of one another (i.e., not bias any samples towards the developed model). Similar patients are identified using a kNN method. For the $k$ matched samples, the GATS technique is employed to combine values of the columns missing in Registry, effectively filling-in the missing features for each Vietnam-based patient. As a result, the HTD and NHTD datasets have a feature set matching the UK data.

Using the comprehensive feature set, we proceeded to perform supplementary experiments by utilizing the OUH pre-pandemic controls and wave one positive cases as the training set, as previously conducted. Subsequently, we re-evaluated the models' performance on the six test sets. Anticipating an enhancement in performance on the UK test sets due to the inclusion of additional features, we also hypothesized that the performance on the Vietnam datasets would also improve (particularly when evaluated using the UK-based models).

We additionally investigate the utility of transfer learning, as this has proven to be a successful approach for applying models developed at one center to another independent center[8]. In our study, we assess the effectiveness of transfer learning by taking the network weights from a trained neural network model, which was initially trained on OUH data. We then fine-tune the network by updating the existing weights using a subset of either the HTD or NHTD data, allowing us to customize the model to the local context of Vietnam. For each of HTD and NHTD, the subset of data selected for transfer learning comprises

the earliest 40% of patients, with 20% used for training and 20% used for threshold adjustment. This allows us to validate the model prospectively on the remaining 60% of patients and externally validate it on the other hospital.

Finally, to establish a baseline, we will train neural network models locally at each hospital in Vietnam. Similar to the transfer learning approach, we will select the earliest 40% of patients from each hospital dataset to train models, with 20% of the data allocated for training and another 20% for threshold adjustment. As before, this setup enables us to perform prospective validation on the remaining 60% of patients within the same hospital and external validation on the dataset from the other hospital (external validation will be performed on the entire dataset).

## Reporting summary
Further information on research design is available in the Nature Portfolio Reporting Summary linked to this article.

## Data availability
Data from OUH studied here are available from the Infections in Oxfordshire Research Database (https://oxfordbrc.nihr.ac.uk/research-themes/modernizing-medical-microbiology-and-big-infection-diagnostics/infections-in-oxfordshire-research-database-iord/), subject to an application meeting the ethical and governance requirements of the Database. Data from UHB, PUH and BH are available by direct request to the hospitals, subject to HRA and research & governance approvals at the individual Trusts. These raw datasets are protected and are not publicly available due to data privacy regulations. Data from HTD and NHTD are available through a managed access policy at OUCRU, through the CCAA Vietnam Data Access Committee, subject to an application meeting the ethical and governance requirements. The data sharing policy can be found here: https://www.oucru.org/data-sharing-policy/. These raw datasets are protected and are not publicly available due to data privacy regulations. Please contact Dr. Louise Thwaites (lthwaites@oucru.org) if you would like help accessing the data. Source data for result Tables and Figures are provided with this paper. Source data are provided with this paper.

## Code availability
Code can be found in: https://github.com/yangjenny/standard_algorithms(https://doi.org/10.5281/zenodo.12789225)[31]

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

## Acknowledgements

This work was supported by the Wellcome Trust/University of Oxford Medical & Life Sciences Translational Fund (Award: 0009350), and the Oxford National Institute for Health and Care Research (NIHR) Biomedical Research Centre (BRC). This work was also supported by the Wellcome Trust (Awards: WT 214906/Z/18/Z and WT217650/Z/19/Z). J.Y. is a Marie Sklodowska-Curie Fellow, under the European Union's Horizon 2020 research and innovation program (Grant agreement: 955681, MOIRA). A.A.S. is an NIHR Academic Clinical Fellow (Award: ACF-2020-13-015). D.A.C. was supported by a Royal Academy of Engineering Research Chair, an NIHR Research Professorship, the InnoHK Hong Kong Centre for Cerebro-cardiovascular Health Engineering (COCHE), and the Pandemic Sciences Institute at the University of Oxford. The funders of the study had no role in study design, data collection, data analysis, data interpretation, or writing of the manuscript. The views expressed in this publication are those of the authors and not necessarily those of the funders. We express our sincere thanks to all patients and staff across the four participating NHS trusts (Oxford University Hospitals NHS Foundation Trust, University Hospitals Birmingham NHS Trust, Bedfordshire Hospitals NHS Foundations Trust, and Portsmouth Hospitals University NHS Trust) and across the two participating Vietnam hospitals (Hospital for Tropical Diseases and the National Hospital for Tropical Diseases). We also express our thanks to the Critical Care Asia Africa Registry team.

## Author contributions

J.Y. conceived and ran the experiments. J.Y. wrote and implemented the code. J.Y. wrote the initial manuscript draft. L.T. and J.Y. applied for data sharing of the Vietnam (HTD and NHTD) data. A.A.S. applied for the ethical approval and co-ordinated data extraction for the UK (OUH, PUH, UHB, BH) data. J.Y. preprocessed the Vietnam datasets. J.Y. and AAS preprocessed the UK COVID-19 datasets. All authors revised the manuscript.

## Competing interests

All authors declare no competing interests.
