## [Peer Review File · Nature Communications]

Reviewers' Comments:

Reviewer #1:

Remarks to the Author:

This paper studies an important topic on investigating the utility of machine learning models trained using data from well-resourced institutions in low-resourced institutions. The current manuscript is immature and I have the following concerns.

1. COVID screening is tricky. There are many existing publications. The authors need to make an overall diagram explaining the design of the entire study. A detailed inclusion exclusion cascade is needed other than Table 1. We also need justification why deploying such a model in low-income countries is desired given there are many such models outside. For example, the one from the following publication

Yang, He S., Yu Hou, Ljiljana V. Vasovic, Peter AD Steel, Amy Chadburn, Sabrina E. Racine-Brzostek, Priya Velu et al. "Routine laboratory blood tests predict SARS-CoV-2 infection using machine learning." *Clinical chemistry* 66, no. 11 (2020): 1396-1404.

2. The feature set the authors used for collecting inputs is neither complicated nor high-dimensional. It is not clear the need or benefit of trying feedforward neural network here. It can only overfit without giving you much improvements over linear models.

3. It is not true that GBDT cannot be easily fine-tuned. Please see the following reference

Yang, He S., Weishen Pan, Yingheng Wang, Mark A. Zaydman, Nicholas C. Spies, Zhen Zhao, Theresa A. Guise, Qing H. Meng, and Fei Wang. "Generalizability of a Machine Learning Model for Improving Utilization of Parathyroid Hormone-Related Peptide Testing across Multiple Clinical Centers." *Clinical chemistry* 69, no. 11 (2023): 1260-1269.

4. It is unfortunate that only retrospective analysis is conducted in the manuscript, though prospective analysis would be much better.

Reviewer #2:

Remarks to the Author:

This paper takes a Covid test-positive prediction model trained in the UK and applies it to a dataset from Vietnam, where it performs less well but still ok. There is a lot of enthusiasm for this general idea (training in a data-rich environment, applying to a less

well-resourced environment) and it's great to see more data on this.

A global comment is that the paper seems quite long, and a bit confusing, for a simple fact. In addition to dramatically cutting words and extraneous details, a diagram of the datasets - where the model was trained, applied, etc. in the various iterations of modeling the authors do - would be very helpful. I'd also remove or replace the many acronyms for the datasets with intelligible words.

I have a couple of technical comments.

First, the authors spend very little time discussing how the Vietnam data were labeled as positive or negative. It seems like this is just using administrative/ICD codes? Were the hospitals testing for Covid during the time the dataset was made? What kinds of tests were they using? Do the authors have any test results to validate the diagnoses, even for a subset of patients? This is important because the labeling is quite important for interpreting the results. If the diagnoses come mostly from physician judgment, not actual test results, I would guess this would make a model based on contextual data (the same contextual data used by physicians - age, etc.) look better than it actually is. I'd appreciate a discussion of this.

Second, the authors use knn to impute a lot of variables in the Vietnam dataset. I don't think there's anything wrong with this, in terms of overfitting, but it is a bit awkward. This is not adding any information in Vietnam - the new results are completely redundant with the other data in Vietnam - so it would be a bit strange if this is improving the model performance much. Also, this is very much part of the modeling step, not the "data cleaning" step: the authors are using information from the UK to impute lab results to patients in Vietnam, which means if someone wanted to apply the authors' model, they would need access to the full UK data (to do knn). That seems like quite a restrictive limitation, unless I'm misunderstanding something?

Reviewer #3:

None

Incorporating Reviewer Feedback:

Generalizability Assessment of AI Models Across Hospitals: A Comparative Study in Low-Middle Income and High Income Countries

Jenny Yang, Nguyen Thanh Dung, Pham Ngoc Thach, Nguyen Thanh Phong, Vu Dinh Phu, Khiem Dong Phu, Lam Minh Yen, Doan Bui Xuan Thy, Andrew A. S. Soltan, Louise Thwaites & David A. Clifton

We are grateful for the time the reviewers have taken to provide comments, which have helped improve and strengthen this manuscript. We have revised the manuscript to incorporate comments, advice, and suggestions. Please find our responses attached below.

Reviewer #1 (Remarks to the Author):

Thank you for the helpful overview and thoughtful insights mentioned/suggested. We are very grateful for the input, and respond below discussing how each point is incorporated. The points mentioned have greatly strengthened and clarified the manuscript.

This paper studies an important topic on investigating the utility of machine learning models trained using data from well-resourced institutions in low-resourced institutions. The current manuscript is immature and I have the following concerns.

1. COVID screening is tricky. There are many existing publications. The authors need to make an overall diagram explaining the design of the entire study. A detailed inclusion exclusion cascade is needed other than Table 1. We also need justification why deploying such a model in low-income countries is desired given there are many such models outside. For example, the one from the following publication

Yang, He S., Yu Hou, Ljiljana V. Vasovic, Peter AD Steel, Amy Chadburn, Sabrina E. Racine-Brzostek, Priya Velu et al. "Routine laboratory blood tests predict SARS-CoV-2 infection using machine learning." *Clinical chemistry* 66, no. 11 (2020): 1396-1404.

We agree that it is really important to emphasize the importance of these models in a LMIC country (particularly in contrast to a HIC).

To highlight key differences between these, we've written in the Introduction:

- LMIC hospitals often face resource constraints, such as inadequate funding, outdated infrastructure, and shortages of technical expertise [7,17]. Additionally, AI algorithms typically rely on large and high-quality datasets for training and validation. However, LMIC hospitals may have limited access to comprehensive and digitized healthcare [1,5,17]. These resource limitations pose significant challenges for the adoption and implementation of healthcare AI systems, especially when compared to many HIC hospitals.

We also mention how this tool has been successful when implemented within a hospital in the UK, and highlighted how these improvements could particularly be beneficial for use in an LMIC setting. Thus, in the Introduction, we mention:

- In the UK, the NHS utilized a green-amber-blue categorization system, where green indicated patients with no COVID-19 symptoms, amber indicated patients with potential COVID-19

symptoms, and blue indicated laboratory-confirmed COVID-19 cases. Through a validation study conducted at the John Radcliffe Hospital in Oxford, England, we demonstrated that our AI screening model improved the sensitivity of lateral flow device (LFD) testing by approximately 30%, and correctly excluded 58.5% of negative patients who were initially triaged as "COVID-19-suspected" by clinicians [13]. Furthermore, the AI model provided diagnoses, on average (median), 16 minutes (26.3%) earlier than LFDs, and 6 hours and 52 minutes (90.2%) earlier than Polymerase Chain Assay (PCR) testing, when the model predictors were collected using point of care full blood count (FBC) analysis. Applying a similar screening tool at the HTD and NHTD in Vietnam could offer a systematic approach to prioritize and manage patient care. It would allow for the efficient use of limited resources, including clinician expertise, ventilators, and beds, ultimately optimizing patient outcomes and ensuring timely access to appropriate interventions. These benefits are especially valuable in LMIC settings where resource constraints pose significant challenges to healthcare delivery (**Yang et al., 2020).

We additionally cite (***) the article mentioned by the reviewer, which highlights the advantage of such a model when sites are faced with supply constraints.

We've also added further information on the inclusion and exclusion criteria, both through descriptions and as a flowchart in the Supplementary Material:

- Oxford University Hospitals NHS Foundation Trust (OUH):
We included all patients attending acute and emergency care settings at OUH who received routine blood tests on arrival, considering presentations before December 1, 2019, and thus before the pandemic, as the COVID-19-negative (control) cohort. We considered presentations during the 'first wave' of the UK COVID-19 pandemic (December 1, 2019 to June 30, 2020) with PCR confirmed SARS-CoV-2 infection as the COVID-19-positive (cases) cohort. We excluded patients who opted out of electronic health record (EHR) research and those who did not receive laboratory blood tests or were younger than 18 years of age. Due to incomplete penetrance of testing during the first wave of the pandemic, and imperfect sensitivity of the PCR test, there is uncertainty in the viral status of patients presenting during the pandemic who were untested or tested negative. We therefore selected a pre-pandemic control cohort during training to ensure absence of disease in patients labelled as COVID-19-negative. Clinical features extracted for each presentation included first-performed blood tests, blood gases, vital signs measurements and PCR testing for SARS-CoV-2 (Abbott Architect [Abbott, Maidenhead, UK], TaqPath [Thermo Fisher Scientific, Massachusetts, USA] and Public Health England-designed RNA-dependent RNA polymerase assays).
- Portsmouth Hospitals University NHS Foundation Trust (PUH):
PUH considered all patients admitted to the Queen Alexandra Hospital, serving a population of 675,000 and offering tertiary referral services to the surrounding region, between March 1, 2020 and February 28, 2021. Confirmatory COVID-19 testing was by laboratory SARS-CoV2 RT-PCR assay, considering any positive PCR result within 48hrs of admission as a true positive.
- University Hospitals Birmingham NHS Foundation Trust (UHB):
UHB considered all patients admitted to The Queen Elizabeth Hospital, Birmingham, between December 01, 2019 and October 29, 2020. The Queen Elizabeth Hospital is a large

tertiary referral unit within the UHB group which provides healthcare services for a population of 2.2 million across the West Midlands. Confirmatory COVID-19 testing was performed by laboratory SARS-CoV-2 RT-PCR assay.

- Bedfordshire NHS Foundation Trust (BH):**
 BH considered all patients admitted to Bedford Hospital between January 1, 2021 and March 31, 2021. BH provides healthcare services for a population of around 620,000 in Bedfordshire. Confirmatory COVID-19 testing was performed on the day of admission by point-of-care PCR based nucleic acid testing [SAMBA-II \& Panther Fusion System, Diagnostics in the Real World, UK, and Hologic, USA].
- Hospital for Tropical Diseases (HTD):**
 HTD considered all patients admitted between January 1, 2021 and December 31, 2022. Confirmatory COVID-19 testing was performed using PCR
- National Hospital for Tropical Diseases (NHTD):**
 NHTD considered all patients admitted between January 1, 2021 and December 31, 2022. Confirmatory COVID-19 testing was performed using PCR and/or rapid antigen testing.

2. The feature set the authors used for collecting inputs is neither complicated nor high-dimensional. It is not clear the need or benefit of trying feedforward neural network here. It can only overfit without giving you much improvements over linear models.

We agree that linear regression and XGBoost models typically provide strong performance across tabular data. Thus, we ensured to present results for linear regression and XGBoost (ensuring we had strong baseline models), alongside neural networks. We wanted to include neural networks as they had also been shown to have strong performance on the UK datasets (enabling fair comparison with other published literature). Additionally, neural networks form the baseline of many other more complex architectures, so demonstrating this can provide insight into other potential architectures which practitioners may be interested in. We made sure to use a MLP with just one hidden layer, to keep it simple (and avoid overfitting that can occur with much more complex deep neural networks). Additionally, neural networks provided a standard and straightforward method for demonstrating the effectiveness of transfer learning. We try to elaborate on this in the Methods section and also additionally highlight the point within the Discussion section:

Section 2.4 Model Architectures:

- In order to evaluate the generalizability of developed models, we conducted investigations using three commonly used model architectures: logistic regression, XGBoost, and a standard neural network. Logistic regression is a linear model that is widely accepted in the clinical community; XGBoost is a tree-based model known for its strong performance on tabular data (**Yang et al., 2023); and lastly, a standard neural network serves as the foundation for many powerful machine learning models and can be used alongside transfer learning. It should be noted that LR is a relatively simple and linear classification model which does not inherently involve complex neural network architectures or deep learning, and thus, is not typically used alongside transfer learning; and XGBoost depends on the availability of the entire dataset, such that transfer learning is not typically feasible [20]. Additionally, a neural network has previously been shown to have superior performance for COVID-19 diagnosis (using the same UK cohorts) [20-23].

Discussion:

- In our research, we consistently observed superior performance of the neural network model when applied to the Vietnam datasets. Nonetheless, it's important to recognize the tendency of neural networks to overfit. Despite employing a straightforward network architecture with only one hidden layer, the risk of overfitting increases when training data is limited, potentially resulting in poor generalization. Hence, it remains crucial to assess simpler models like Logistic Regression (LR) and XGBoost as benchmarks, as demonstrated in our analysis.

3. It is not true that GBDT cannot be easily fine-tuned. Please see the following reference

Yang, He S., Weishen Pan, Yingheng Wang, Mark A. Zaydman, Nicholas C. Spies, Zhen Zhao, Theresa A. Guise, Qing H. Meng, and Fei Wang. "Generalizability of a Machine Learning Model for Improving Utilization of Parathyroid Hormone-Related Peptide Testing across Multiple Clinical Centers." *Clinical chemistry* 69, no. 11 (2023): 1260-1269.

We agree that a GBDT can be fine-tuned with respect to parameters; however, we have specifically focused on the idea of transfer learning, which GBDT typically isn't used for (as GBDT takes the entire dataset to determine the tree-based decision-making process).

Thank you for mentioning the reference, as it does highlight the strength of XGBoost using similar types of features. Thus, we have cited (***) it in the paper.

These can be seen together in the response to point 2.

4. It is unfortunate that only retrospective analysis is conducted in the manuscript, though prospective analysis would be much better.

Yes, this is a good point, however we were limited in which data were available to us during the time of the study. We agree that prospective analysis would be ideal, and add this point to the Discussion, highlighting it as an important future direction.

Discussion:

- Finally, our investigation spanned a significant time period, from December 1, 2019, to December 30, 2022. During this extended duration and particularly during peak pandemic periods, such as the COVID-19 outbreak, the relationship between patient and disease factors with clinical events, including hospital-acquired infections, may undergo changes [6]. Additionally, over time, there may be variations in practice patterns such as hardware and software updates and changes in protocols, which can impact data capture and outcomes. While this retrospective study provided valuable insights into historical data, ideally, future research should prioritize prospective analysis, which allows for a more dynamic assessment of model performance and can provide timely feedback for refining and improving predictive models. Therefore, in future works, emphasis should be placed on implementing prospective analysis to enhance the robustness and applicability of the findings.

Reviewer #2 (Remarks to the Author):

Thank you for the time you've taken to review our manuscript and for the points you've listed, as they've helped enhance the takeaway message of our paper. We are very grateful for the input and respond below with how each point is incorporated.

This paper takes a Covid test-positive prediction model trained in the UK and applies it to a dataset from Vietnam, where it performs less well but still ok. There is a lot of enthusiasm for this general idea (training in a data-rich environment, applying to a less well-resourced environment) and it's great to see more data on this.

A global comment is that the paper seems quite long, and a bit confusing, for a simple fact. In addition to dramatically cutting words and extraneous details, a diagram of the datasets - where the model was trained, applied, etc. in the various iterations of modeling the authors do - would be very helpful. I'd also remove or replace the many acronyms for the datasets with intelligible words.

We acknowledge the potential confusion that can arise from excessive use of acronyms. Regarding the naming of different hospitals, we made the decision to retain the acronyms since these hospital names are frequently referenced throughout the manuscript and are notably lengthy, such as University Hospitals Birmingham NHS Trust, abbreviated as UHB. This choice was made to balance considerations of both word count and readability. Additionally, these maintain consistency with the acronyms used in other studies that utilized the same UK sets. However, we made efforts to minimize the use of other acronyms where possible, opting for descriptive terms like "emergency department" instead of "ED" when clarity allowed.

I have a couple of technical comments.

First, the authors spend very little time discussing how the Vietnam data were labeled as positive or negative. It seems like this is just using administrative/ICD codes? Were the hospitals testing for Covid during the time the dataset was made? What kinds of tests were they using? Do the authors have any test results to validate the diagnoses, even for a subset of patients? This is important because the labeling is quite important for interpreting the results. If the diagnoses come mostly from physician judgment, not actual test results, I would guess this would make a model based on contextual data (the same contextual data used by physicians - age, etc.) look better than it actually is. I'd appreciate a discussion of this.

We have clarified this and included more details on how COVID-19 was labeled at each site. We additionally performed sensitivity analysis to address the uncertainty surrounding the viral status of patients who underwent rapid antigen testing or where the testing method was unspecified.

Section 2.1 Data:

- We then validated the model on four UK cohorts (OUH “wave 2”, UHB, PUH, BH), totalling 72,223 admitted patients (4,600 COVID-19 positive), and two Vietnam cohorts (HTD and NHTD), totalling 3,431 admitted patients (2,413 COVID-19 positive). A summary of each respective cohort is in Table 1. Full inclusion and exclusion criteria are provided in the Supplementary Material.

For OUH, we included all patients presenting and admitted to the ED. For PUH, UHB, BH, HTD, and NHTD, we included all patients admitted to the ED. COVID-19 status at the UK sites and HTD was determined through confirmatory PCR testing, while at NHTD, both PCR and/or rapid antigen testing were used. Nonetheless, concerning NHTD, there were numerous instances where the specific test type was not recorded. Therefore, in order to maximize testing coverage, in cases where the test type was unspecified, we examined how COVID-19 was documented, including terms such as "COVID-19 lower respiratory infection," "COVID-19 pneumonia," "SARS-COV-2 Infection," "COVID-19 acute respiratory distress syndrome," "Acute COVID-19," and others. For our analysis, alongside confirmatory testing, we considered any indication and severity of COVID-19 presence as COVID-19 positive. These diagnoses were confirmed by attending specialist infectious diseases clinicians, and thus, we consider these diagnostic labels to be robust. A detailed breakdown of the labels available within the NHTD database is provided in Supplementary Table 1. Furthermore, we conducted a sensitivity analysis for NHTD, comparing PCR-confirmed outcomes with those incorporating rapid antigen tests and other written documentation of COVID-19, which is detailed in Section 3.

Section 3 Results:

- We conducted an additional sensitivity analysis to address the uncertainty surrounding the viral status of patients who underwent rapid antigen testing or where the testing method was unspecified at NHTD. Utilizing the NN model, which demonstrated superior performance, and evaluating solely on the subset of NHTD patients with confirmed PCR testing, we attained AUROC scores of 0.492 (0.445-0.548) for the NHTD set. Generally, we observed comparable results, indicated by overlapping confidence intervals, when compared to

datasets incorporating alternative testing methods. Consequently, further experiments were conducted using the dataset encompassing testing methods beyond PCR.

We've also added further information on the inclusion and exclusion criteria, both through descriptions and as a flowchart in the Supplementary Material:

- **Oxford University Hospitals NHS Foundation Trust (OUH):**
We included all patients attending acute and emergency care settings at OUH who received routine blood tests on arrival, considering presentations before December 1, 2019, and thus before the pandemic, as the COVID-19-negative (control) cohort. We considered presentations during the 'first wave' of the UK COVID-19 pandemic (December 1, 2019 to June 30, 2020) with PCR confirmed SARS-CoV-2 infection as the COVID-19-positive (cases) cohort. We excluded patients who opted out of electronic health record (EHR) research and those who did not receive laboratory blood tests or were younger than 18 years of age. Due to incomplete penetrance of testing during the first wave of the pandemic, and imperfect sensitivity of the PCR test, there is uncertainty in the viral status of patients presenting during the pandemic who were untested or tested negative. We therefore selected a pre-pandemic control cohort during training to ensure absence of disease in patients labelled as COVID-19-negative. Clinical features extracted for each presentation included first-performed blood tests, blood gases, vital signs measurements and PCR testing for SARS-CoV-2 (Abbott Architect [Abbott, Maidenhead, UK], TaqPath [Thermo Fisher Scientific, Massachusetts, USA] and Public Health England-designed RNA-dependent RNA polymerase assays).
- **Portsmouth Hospitals University NHS Foundation Trust (PUH):**
PUH considered all patients admitted to the Queen Alexandra Hospital, serving a population of 675,000 and offering tertiary referral services to the surrounding region, between March 1, 2020 and February 28, 2021. Confirmatory COVID-19 testing was by laboratory SARS-CoV2 RT-PCR assay, considering any positive PCR result within 48hrs of admission as a true positive.
- **University Hospitals Birmingham NHS Foundation Trust (UHB):**
UHB considered all patients admitted to The Queen Elizabeth Hospital, Birmingham, between December 01, 2019 and October 29, 2020. The Queen Elizabeth Hospital is a large tertiary referral unit within the UHB group which provides healthcare services for a population of 2.2 million across the West Midlands. Confirmatory COVID-19 testing was performed by laboratory SARS-CoV-2 RT-PCR assay.
- **Bedfordshire NHS Foundation Trust (BH):**
BH considered all patients admitted to Bedford Hospital between January 1, 2021 and March 31, 2021. BH provides healthcare services for a population of around 620,000 in Bedfordshire. Confirmatory COVID-19 testing was performed on the day of admission by point-of-care PCR based nucleic acid testing [SAMBA-II \& Panther Fusion System, Diagnostics in the Real World, UK, and Hologic, USA].
- **Hospital for Tropical Diseases (HTD):**
HTD considered all patients admitted between January 1, 2021 and December 31, 2022. Confirmatory COVID-19 testing was performed using PCR

- National Hospital for Tropical Diseases (NHTD):
 NHTD considered all patients admitted between January 1, 2021 and December 31, 2022.
 Confirmatory COVID-19 testing was performed using PCR and/or rapid antigen testing.

Second, the authors use knn to impute a lot of variables in the Vietnam dataset. I don't think there's anything wrong with this, in terms of overfitting, but it is a bit awkward. This is not adding any information in Vietnam - the new results are completely redundant with the other data in Vietnam - so it would be a bit strange if this is improving the model performance much. Also, this is very much part of the modeling step, not the "data cleaning" step: the authors are using information from the UK to impute lab results to patients in Vietnam, which means if someone wanted to apply the authors' model, they would need access to the full UK data (to do knn). That seems like quite a restrictive limitation, unless I'm misunderstanding something?

The approach employed combines kNN with the utilization of convex combinations of complete data. Given the potential challenges of incomplete data in LMIC datasets, the inclusion of HIC data, which tends to be more comprehensive, can serve to enhance or fill gaps in LMIC data. We present a specific method, termed GATS, wherein missing data can be substituted with plausible values that closely approximate the true ones, drawing from insights gained from the complete UK data. While we are particularly emphasizing the challenge of constructing models in HIC settings for use in LMICs, access to UK data isn't the primary concern, as that's where the model originates. Given the manuscript's primary focus on generalizability, we aim to demonstrate how filling in missing data through such a method could potentially improve the generalizability of HIC-trained models to LMIC settings. We agree that this proposed method is more of a data augmentation/pre-processing step rather than a cleaning step. We try to clarify this proposed method in the following sections:

Section 2.6 Training Outline:

- In their study, Soltan et al. [12] identified specific laboratory blood markers, such as eosinophils and basophils, as having a significant impact on model predictions. This determination was made through the application of SHAP (SHapley Additive exPlanations) analysis during the development and evaluation of their models using patient cohorts from the UK. However, these particular features were not accessible in the Registry dataset, and consequently, were not incorporated into the initial models developed for compatible testing across UK and Vietnam cohorts. We hypothesize that without the inclusion of these features during training, the models' performance would be inferior compared to the previously reported scores. Hence, our goal is to illustrate how HICs could potentially assist LMICs by facilitating dataset augmentation or completion, which can improve outcomes when applying an HIC model to an LMIC setting.

In the context of addressing missing data, nearest neighbor (NN) imputation algorithms provide efficient approaches for completing missing values. In these methods, each absent value in certain records gets replaced by a value derived from related cases within the entire dataset [2]. This approach has the capacity to substitute missing data with plausible values that closely approximate the true ones.

Drawing from a similar concept, a recent technique called "Geometrically-Aggregated Training Samples (GATS)" [25] has been introduced to address missing data challenges. GATS constructs training samples by blending various patient characteristics using convex combinations.

Discussion:

- Furthermore, it's important to recognize that while HICs typically possess extensive collections of health data, many LMICs face limitations in data availability, particularly regarding the volume and quality of data accessible electronically and the asynchronous, varied nature of information. These factors can make it challenging to train AI models [5,9] Therefore, in the context of LMICs, where datasets may be smaller and data accessibility issues persist, it is advisable to consider additional computational techniques such as "GATS" to enhance the utilization of available data resources and ultimately improve the effectiveness and generalizability of HIC-trained models in LMIC settings.

Reviewers' Comments:

Reviewer #1:

Remarks to the Author:

Thanks for addressing my concerns carefully, I do not have further comments.

Reviewer #3:

Remarks to the Author:

The authors meticulously addressed the issues raised by both reviewers in the best possible way given the limitations of the datasets. Claims were appropriately toned down without discounting the contribution of the work.

There are a few suggestions that would strengthen the manuscript even more.

1. Some exploration of the errors of the models should be required when reporting performance of machine learning models. This will take the place of fairness evaluation which is glaringly missing, which most likely stems from the lack of demographic labels in the datasets. But that should not hinder error interrogation. Are there similarities among the false negatives, and how do they differ from the true negatives? In the same token, are there commonalities among the false positives, and how do they differ from the true positives? Error interrogation may help identify patients who are most likely going to be harmed by artificial intelligence.
2. The acknowledgement that a prospective evaluation is the requisite next step is appreciated (if not expected). Can the authors elaborate on how they foresee data preprocessing and curation to occur in real-time? What happens if the algorithm is run on raw data prior to the removal of outliers, imputation, etc.? This seems an important intermediate step prior to the prospective evaluation that is perfunctorily added at the end of almost every machine learning in health manuscript and submission. Is it possible to run this simple experiment?
3. There is evidence that statements such as "Data are available on reasonable request" mean nothing and perhaps should not be banned in publications.
4. The title, "A Comparative Study in Low-Middle Income and High Income Countries" seems a bit of a stretch given that the study involved the UK and Vietnam.

Incorporating Reviewer Feedback:

Generalizability Assessment of AI Models Across Hospitals: A Comparative Study in a Low-Middle Income and a High Income Country

Jenny Yang, Nguyen Thanh Dung, Pham Ngoc Thach, Nguyen Thanh Phong, Vu Dinh Phu, Khiem Dong Phu, Lam Minh Yen, Doan Bui Xuan Thy, Andrew A. S. Soltan, Louise Thwaites & David A. Clifton

Reviewer #3 (Remarks to the Author):

Thank you for the helpful overview and thoughtful insights mentioned/suggested. We are very grateful for the input, and respond below discussing how each point is incorporated.

The authors meticulously addressed the issues raised by both reviewers in the best possible way given the limitations of the datasets. Claims were appropriately toned down without discounting the contribution of the work.

There are a few suggestions that would strengthen the manuscript even more.

1. Some exploration of the errors of the models should be required when reporting performance of machine learning models. This will take the place of fairness evaluation which is glaringly missing, which most likely stems from the lack of demographic labels in the datasets. But that should not hinder error interrogation. Are there similarities among the false negatives, and how do they differ from the true negatives? In the same token, are there commonalities among the false positives, and how do they differ from the true positives? Error interrogation may help identify patients who are most likely going to be harmed by artificial intelligence.

We have added error analysis in both the Results section, and also commented on this in the Discussion. Full plots of feature distributions across subgroups and classes is also provided in the Supplementary Material.:

- Results: “We conducted a subgroup analysis for both correct and incorrect classifications across COVID-19-negative and COVID-19-positive groups, focusing on the features used for prediction. This was evaluated on the neural network model. Patients with lower white blood cell counts exhibited higher false negative rates at both HTD and NHTD. At NHTD, haemoglobin and platelet values also showed notable differences in distribution regarding false positive and false negative rates. Detailed prediction distribution plots by class are available in Section D of the Supplementary Material.”
- Discussion: “Furthermore, as we conducted error analysis on subgroups related to classification features, it is imperative for future research to extend this analysis to demographic subgroups. This would enhance our comprehension of how various groups might experience differential impacts from a machine learning algorithm (we did not have these features fully available in the datasets, and thus were not able to perform these analyses).”

2. The acknowledgement that a prospective evaluation is the requisite next step is appreciated (if not expected). Can the authors elaborate on how they foresee data preprocessing and curation to occur in real-time?

We have elaborated on this in the Discussion:

- “While this retrospective study provided valuable insights into historical data, future research should ideally prioritize prospective analysis. This approach allows for a more dynamic assessment of model performance and provides timely feedback for refining and improving predictive models. Therefore, future work should emphasize implementing prospective analysis to enhance the robustness and applicability of the findings. For instance, in LMIC settings, real-time data preprocessing and curation can be achieved through cost-effective and accessible strategies. An offline, in-house version of the algorithm can be used, where a doctor manually enters feature values in real-time (feasible with only 14 features). These values are then automatically processed through a script that imputes missing features and performs standardization, ultimately outputting a diagnosis for further triaging. Additionally, emphasizing the use of open-source tools and scalable, cost-effective infrastructure ensures applicability in resource-constrained settings”

What happens if the algorithm is run on raw data prior to the removal of outliers, imputation, etc.? This seems an important intermediate step prior to the prospective evaluation that is perfunctorily added at the end of almost every machine learning in health manuscript and submission. Is it possible to run this simple experiment?

We did not remove outliers and further discuss this:

- “Regarding data quality, we also detected the presence of outliers within the Vietnam datasets, such as the minimum recorded haemoglobin value of 11 g/L. This particular value would typically be considered highly improbable \cite{beutler,thomaslumb}. The existence of such outliers could be attributed to a unit conversion error, where values were erroneously shifted by a factor of 10 (some locations utilize g/dL instead of g/L), or they may be the result of data entry errors. Since we aimed to work with real data, our model incorporates such instances of incorrect data entry and outliers. In the case of extreme values for white blood cell count, there were some extreme values found in patients with lymphoma in the HTD and NHTD datasets. In certain scenarios, outliers like these may contain unique information that can enhance a model's ability to generalize effectively, rendering the models more robust and less susceptible to noise. The decision of whether to retain extreme values in a dataset or not depends on the context and the problem under consideration. Extreme values can indeed offer valuable information, but it is important to handle them appropriately to prevent any adverse impact on model performance \cite{smiti, tropsha}. Therefore, for future studies, it may be worthwhile to explore additional filtering and preprocessing steps to address these anomalies and enhance the dataset's quality before model development and testing.”

With respect to imputation, we have added the requested experiment and highlight it in the Methods section:

- “We first ensured uniformity in the measurement units for identical features. Next, we standardized all features to have a mean of 0 and a standard deviation of 1 to aid in achieving convergence in neural network models. To address missing values in the UK datasets, we used population median imputation. These steps are consistent with (13 ; 21 ; 22; 23 ; 24). For matched features in the Vietnamese datasets, we also applied population

median imputation. Additionally, we performed sensitivity analysis for these cohorts to account for missing values using the XGBoost model, which is the baseline model from previous studies and can handle missing values as input. Using the test set, we achieved AUROC scores of 0.525 (0.500-0.553) and 0.427 (0.401-0.452) for the HTD and NHTD sets with missing values, respectively. For the imputed sets, the AUROC scores improved to 0.533 (0.508–0.559) and 0.478 (0.453-0.504) for HTD and NHTD, respectively. The technique used to handle completely missing features in the Vietnam datasets is discussed in the following sections.”

3. There is evidence that statements such as "Data are available on reasonable request" mean nothing and perhaps should not be banned in publications.

We've tried to make this more detailed with respect to how data can be accessed.

- Data from OUH studied here are available from the Infections in Oxfordshire Research Database (<https://oxfordbrc.nihr.ac.uk/research-themes/modernising-medical-microbiology-and-big-infection-diagnostics/infections-in-oxfordshire-research-database-iord/>), subject to an application meeting the ethical and governance requirements of the Database. Data from UHB, PUH and BH are available on reasonable request to the respective trusts, subject to HRA requirements. Please contact Dr. Andrew Soltan (andrew.soltan@oncology.ox.ac.uk) if you would like help with accessing the data.

Data from HTD and NHTD are available through a managed access policy at OUCRU, through the CCAA Vietnam Data Access Committee, subject to an application meeting the ethical and governance requirements. The data sharing policy can be found here: (<https://www.oucru.org/data-sharing-policy/>). Please contact Dr. Louise Thwaites (lthwaites@oucru.org) if you would like help accessing the data.

4. The title, "A Comparative Study in Low-Middle Income and High Income Countries" seems a bit of a stretch given that the study involved the UK and Vietnam.

We've changed the title to: **Generalizability Assessment of AI Models Across Hospitals: A Comparative Study in a Low-Middle Income and a High Income Country**

Given that our discussion revolves around algorithmic generalizability across low- and high-income countries, we aim to maintain a broad title that centers on the concept rather than specifically referencing the two countries involved.

Reviewers' Comments:

Reviewer #3:

Remarks to the Author:

Thank you for the opportunity to review the revised manuscript. Full disclosure, I was not involved with the initial review. The authors addressed all the comments and questions of the reviewers adequately. Technically, the manuscript is robust. But prediction models are never truly validated due to expected heterogeneity in model performance between locations, settings, and over time. A single external validation study in a specific geographical location, time frame, and patient population only provides a snapshot and cannot claim universal transportability.

In addition, prediction models have an implicit expiration date and need to be updated regularly to stay relevant. Validation should focus on quantifying and understanding performance heterogeneity, not just assessing point estimates.

The paper also fails to acknowledge the much larger issues that threaten to upend the promise of machine learning to improve care delivery in low-resource settings. The adoption of AI in LMICs presents an ironic situation, as these countries often struggle with basic technological issues like power outages, internet connectivity, and cybersecurity. We have been increasingly dubious of the impact of academic institutions from HICs investing in co-developing AI solutions with colleagues in LMICs. The lack of stable infrastructure and skilled professionals in the AI sector poses significant challenges for the meaningful implementation of AI in many LMICs. Most LMICs face issues such as lack of data and inadequate digital infrastructure, which undermine the potential for AI applications, and the focus on AI in LMICs may be distracting from more pressing systems and technological needs and priorities. Even HIC efforts to extend AI-related policies and guidelines to LMICs may be viewed as a form of "hegemonizing tendency" by tech-savvy countries.

The philosophical underpinnings of the AI discourse might seem beyond the scope of the paper, but we must remind the machine learning community every opportunity that we get that when we build models, we often miss the forest for the trees. For example, AI inequities are frequently framed as bias, seen as a sort of technical issue, where fairness is perceived as some kind of statistical parity. But behind the discourse of scientific and technological neutrality comes a series of normative judgements.

We also suggest at least mentioning the regulatory quagmire around AI. Even well-resourced academic medical centers in HICs are struggling to effectively govern predictive AI tools, indicating a need for more guidance and oversight beyond individual health systems. If well-resourced academic centres are struggling to oversee AI

deployment, there is little hope for critical access hospitals in HICs and most if not all health systems in LMICs.

Lastly, there are muted callouts how the interest and investments in AI are siphoning funding from more pressing systems and technological needs and priorities, such as improving basic services and infrastructure.

References:

1. Van Calster, B., Steyerberg, E.W., Wynants, L. et al. There is no such thing as a validated prediction model. *BMC Med* 21, 70 (2023). <https://doi.org/10.1186/s12916-023-02779-w>
2. Biana, H.T., Joaquin, J.J. The irony of AI in a low-to-middle-income country. *AI & Soc* (2024). <https://doi.org/10.1007/s00146-023-01855-2>
3. Nong P, Hamasha R, Singh K, Adler-Milstein J, Platt J. How academic medical centers govern AI prediction tools in the context of uncertainty and evolving Rregulation. *NEJM AI*. 2024 Jan 31;1(3). doi: 10.1056/Alp2300048
4. Sekalala S, Chatikobo T. Colonialism in the new digital health agenda. *BMJ Glob Health*. 2024 Feb 27;9(2):e014131. doi: 10.1136/bmjgh-2023-014131. PMID: 38413105; PMCID: PMC10900325.
5. Futoma J, Simons M, Panch T, Doshi-Velez F, Celi LA. The myth of generalisability in clinical research and machine learning in health care. *Lancet Digit Health*. 2020 Sep;2(9):e489-e492. doi: 10.1016/S2589-7500(20)30186-2. Epub 2020 Aug 24. PMID: 32864600; PMCID: PMC7444947.

Incorporating Reviewer Feedback:

Generalizability Assessment of AI Models Across Hospitals in a Low-Middle and High Income Country

Jenny Yang, Nguyen Thanh Dung, Pham Ngoc Thach, Nguyen Thanh Phong, Vu Dinh Phu, Khiem Dong Phu, Lam Minh Yen, Doan Bui Xuan Thy, Andrew A. S. Soltan, Louise Thwaites & David A. Clifton

Reviewer #3 (Remarks to the Author):

We thank the reviewer for taking the time to review our revised manuscript and provide further comments. We respond to the suggestions below.

Thank you for the opportunity to review the revised manuscript. Full disclosure, I was not involved with the initial review. The authors addressed all the comments and questions of the reviewers adequately. Technically, the manuscript is robust. But prediction models are never truly validated due to expected heterogeneity in model performance between locations, settings, and over time. A single external validation study in a specific geographical location, time frame, and patient population only provides a snapshot and cannot claim universal transportability.

In addition, prediction models have an implicit expiration date and need to be updated regularly to stay relevant. Validation should focus on quantifying and understanding performance heterogeneity, not just assessing point estimates.

This is mentioned in the Discussion:

- It is also important to acknowledge that prediction models can never be fully validated due to inherent variability in their performance across different locations, settings, and time periods \cite{vancalster, futoma}. A single external validation study conducted in a specific geographical area, during a particular time frame, and within a distinct patient population offers only a limited view and cannot assert universal applicability. In this study, our investigation spanned a significant time period, from December 1, 2019, to December 30, 2022. During this extended duration and particularly during peak pandemic periods, such as the COVID-19 outbreak, the relationship between patient and disease factors with clinical events, including hospital-acquired infections, may undergo changes \cite{futoma}. Additionally, over time, there may be variations in practice patterns such as hardware and software updates and changes in protocols, which can impact data capture and outcomes. Although this retrospective study offered valuable insights into historical data, future research should ideally focus on prospective analysis. Models should be updated regularly to maintain their relevance. This approach enables a more dynamic assessment of model performance and provides timely feedback for refining and improving predictive models.

Therefore, future efforts should validation efforts should aim to quantify and comprehend the heterogeneity in model performance, rather than solely focusing on point estimates \cite{vancalster}. This broader understanding of performance variability is crucial for refining and improving the models over time. For instance, in LMIC settings, real-time data preprocessing and curation can be achieved through cost-effective and accessible strategies. In the study highlighted here, an offline, in-house version of the algorithm can be used, where a doctor manually enters feature values in real-time (feasible with only 14 features). These values can then be automatically processed through a script that imputes missing features and performs standardization, ultimately outputting a diagnosis for further triaging. Additionally, emphasizing the use of open-source tools and scalable, cost-effective infrastructure ensures applicability in resource-constrained settings.

The paper also fails to acknowledge the much larger issues that threaten to upend the promise of machine learning to improve care delivery in low-resource settings. The adoption of AI in LMICs presents an ironic situation, as these countries often struggle with basic technological issues like power outages, internet connectivity, and cybersecurity. We have been increasingly dubious of the impact of academic institutions from HICs investing in co-developing AI solutions with colleagues in LMICs. The lack of stable infrastructure and skilled professionals in the AI sector poses significant challenges for the meaningful implementation of AI in many LMICs. Most LMICs face issues such as lack of data and inadequate digital infrastructure, which undermine the potential for AI applications, and the focus on AI in LMICs may be distracting from more pressing systems and technological needs and priorities. Even HIC efforts to extend AI-related policies and guidelines to LMICs may be viewed as a form of "hegemonizing tendency" by tech-savvy countries.

The philosophical underpinnings of the AI discourse might seem beyond the scope of the paper, but we must remind the machine learning community every opportunity that we get that when we build models, we often miss the forest for the trees. For example, AI inequities are frequently framed as bias, seen as a sort of technical issue, where fairness is perceived as some kind of statistical parity. But behind the discourse of scientific and technological neutrality comes a series of normative judgements.

We also suggest at least mentioning the regulatory quagmire around AI. Even well-resourced academic medical centers in HICs are struggling to effectively govern predictive AI tools, indicating a need for more guidance and oversight beyond individual health systems. If well-resourced academic centres are struggling to oversee AI deployment, there is little hope for critical access hospitals in HICs and most if not all health systems in LMICs.

Lastly, there are muted callouts how the interest and investments in AI are siphoning

funding from more pressing systems and technological needs and priorities, such as improving basic services and infrastructure.

We've added this to the Discussion:

- Finally, the adoption of AI in LMICs encounters significant infrastructural and capacity-building challenges \cite{yangbiaslmic, labrique, ciecierski, alami}. These challenges encompass power outages, unreliable internet connectivity, cybersecurity concerns, inadequate digital infrastructure (such as data and storage), and a shortage of skilled AI professionals. As a result, prioritizing AI solutions may divert resources from more urgent foundational needs. These issues also impact the broader concern of AI governance, which remains a challenge even in HICs \cite{nong}, and is likely even more challenging in LMICs. Therefore, while AI holds promise, its adoption in LMICs necessitates a careful, context-sensitive approach to address these underlying challenges.

References:

1. Van Calster, B., Steyerberg, E.W., Wynants, L. et al. There is no such thing as a validated prediction model. *BMC Med* 21, 70 (2023). <https://doi.org/10.1186/s12916-023-02779-w>
2. Biana, H.T., Joaquin, J.J. The irony of AI in a low-to-middle-income country. *AI & Soc* (2024). <https://doi.org/10.1007/s00146-023-01855-2>
3. Nong P, Hamasha R, Singh K, Adler-Milstein J, Platt J. How academic medical centers govern AI prediction tools in the context of uncertainty and evolving Rregulation. *NEJM AI*. 2024 Jan 31;1(3). doi: 10.1056/AIp2300048
4. Sekalala S, Chatikobo T. Colonialism in the new digital health agenda. *BMJ Glob Health*. 2024 Feb 27;9(2):e014131. doi: 10.1136/bmjgh-2023-014131. PMID: 38413105; PMCID: PMC10900325.
5. Futoma J, Simons M, Panch T, Doshi-Velez F, Celi LA. The myth of generalisability in clinical research and machine learning in health care. *Lancet Digit Health*. 2020 Sep;2(9):e489-e492. doi: 10.1016/S2589-7500(20)30186-2. Epub 2020 Aug 24. PMID: 32864600; PMCID: PMC7444947.